# Pyroptosis inhibiting nanobodies block Gasdermin D pore formation

Anja Kopp [1,2], Gregor Hagelueken [1], Isabell Jamitzky [1], Jonas Moecking [1], Lisa D. J. Schiffelers [3], Florian I. Schmidt [3] & Matthias Geyer [1]✉

Human Gasdermin D (GSDMD) is a key mediator of pyroptosis, a pro-inflammatory form of cell death occurring downstream of inflammasome activation as part of the innate immune defence. Upon cleavage by inflammatory caspases in the cytosol, the N-terminal domain of GSDMD forms pores in the plasma membrane resulting in cytokine release and eventually cell death. Targeting GSDMD is an attractive way to dampen inflammation. In this study, six GSDMD targeting nanobodies are characterized in terms of their binding affinity, stability, and effect on GSDMD pore formation. Three of the nanobodies inhibit GSDMD pore formation in a liposome leakage assay, although caspase cleavage was not perturbed. We determine the crystal structure of human GSDMD in complex with two nanobodies at 1.9 Å resolution, providing detailed insights into the GSDMD–nanobody interactions and epitope binding. The pore formation is sterically blocked by one of the nanobodies that binds to the oligomerization interface of the N-terminal domain in the multi-subunit pore assembly. Our biochemical and structural findings provide tools for studying inflammasome biology and build a framework for the design of GSDMD targeting drugs.

The human gasdermin family consists of six differentially expressed members (GSDM A to E and PVJK/GSDMF) that exert diverse functions in inflammation and cell death[1,2]. Gasdermin D (GSDMD) is a cytosolic protein that serves as a key mediator of pyroptosis, a pro-inflammatory form of cell death occurring in the context of microbial infection or tissue damage as part of the innate immune response[3,4]. Sensing of cellular or pathogen-derived danger signals triggers the assembly of canonical and non-canonical inflammasomes, which leads to the activation of inflammatory caspases in the cytoplasm (caspase-1, −4, and −5 in human or caspases-1 and −11 in mice)[5,6]. These caspases were found to cleave GSDMD at a conserved sequence motif (FLTD$_{275}$ | GV in humans, LLSD$_{276}$ | GI in mice), residing in a long linker region between the GSDMD N- and C-terminal domains of the 52.8 kDa protein[3,4]. As in most gasdermins, the N-terminal domain (NTD) is cytotoxic and repressed by the auto-inhibitory C-terminal domain (CTD) in the inactive state[7–10]. Upon caspase cleavage, the two domains can

dissociate by an as yet unknown mechanism, with the NTD capable of forming transmembrane pores. However, the exact mechanisms of NTD release, oligomerization, plasma membrane association, insertion, and conformational changes required to induced pore formation remain elusive[3,4,11,12].

The structure of the GSDMD pore was determined by cryogenic electron microscopy, revealing two different conformational states of the pore: a membrane-associated 'pre-pore' and a mature membrane-spanning pore consisting of approximately 33 NTD subunits with an inner diameter of 21 nm (ref. 13). Recent molecular simulations of the mechanism of GSDMD pore-formation suggest a concentration-dependent process, in which low concentrations of GSDMD NTD at the membrane lead to the formation of small oligomers and sublytic pores that have the potential to grow into larger pores, whereas high concentrations of GSDMD NTD lead to the assembly of larger pre-pores[14]. The pores open and close dynamically dependent on the

[1]Institute of Structural Biology, Medical Faculty, University of Bonn, Venusberg-Campus 1, 53127 Bonn, Germany. [2]Inflammation Division, The Walter and Eliza Hall Institute of Medical Research, Parkville, VIC 3052, Australia. [3]Institute of Innate Immunity, Medical Faculty, University of Bonn, Venusberg-Campus 1, 53127 Bonn, Germany. ✉e-mail: matthias.geyer@uni-bonn.de

phosphoinositide environment in the membrane[15]. GSDMD pores enable the release of the pro-inflammatory cytokines IL-1β and IL-18, which initiates further immune responses[3,4,11,13,16,17]. Water influx through GSDMD pores causes cell swelling and imbalances in the cellular ion homeostasis that, in concert with activation of the protein NINJ1, ultimately result in membrane rupture, the release of cellular contents, and cell death[18–20]. Cytokine secretion can be limited by calcium influx-induced activation of the membrane-remodelling ESCRT-III machinery[21] and the Ragulator-Rag complex has been reported to be necessary for GSDMD oligomerization in the plasma membrane of macrophages[22,23]. However, the exact mechanisms behind GSDMD pore formation and regulatory processes remain to be investigated.

Since GSDMD mediates the final common step of all inflammasome pathways it plays a central role in innate immunity and has been implicated in numerous diseases with aberrant inflammasome activation such as familial Mediterranean fever (FMF), atherosclerosis, inflammatory bowel disease (IBD), gout, Alzheimer's disease and sepsis[24–29]. Therefore, targeting GSDMD with small molecule inhibitors is an attractive strategy to dampen inflammation. Recently, the three cysteine-reactive molecules disulfiram, necrosulfonamide, and dimethyl-fumarate have been identified to covalently modify Cys191 in the GSDMD NTD and inhibit GSDMD pore formation[30–32]. Although these molecules were effective in murine sepsis models, off-target effects must be carefully studied due to the cysteine-reactive nature of these inhibitors.

In this study, six GSDMD targeting nanobodies have been analyzed by biochemical and structural means. Nanobodies are single-domain antibody fragments (also named VHH) derived from the heavy chain only antibodies naturally occurring in camelids[33,34]. Due to their small size, stability, high binding affinities, and low production costs, nanobodies find broad applications as tools in cellular biology, imaging, structural biology, and pharmacology[35,36]. We found that three of the GSDMD targeting nanobodies inhibited GSDMD pore formation in an in vitro liposome leakage assay. We determined the crystal structure of full-length GSDMD in complex with two nanobodies (one inhibitory, one non-inhibitory) at 1.9 Å resolution and characterized the binding epitopes biochemically. Due to their high specificity and binding affinity, the nanobodies can be used as tools to study GSDMD pore formation and inflammasome biology, as crystallization chaperones to determine high-resolution structures and, most importantly, as starting points for the development of GSDMD-specific biological drugs.

## Results

### Identification of GSDMD specific nanobodies

GSDMD targeting nanobodies were raised by immunization of an alpaca with full-length recombinant human GSDMD protein. Identification by phage display and initial characterization of binding analyzed by enzyme-linked immunosorbent assay (ELISA) and LUMIER assays are described by Schiffelers et al.[37]. The potential binders differed by at least 7.6% in their amino acid sequence and showed great variety in the lengths and composition of their complementarity determining region 3 (CDR3) (Fig. 1a, b). The nanobodies as well as human wild-type, full length GSDMD protein were recombinantly expressed in E. coli cells and displayed by affinity purification followed by size exclusion chromatography (Supplementary Fig. 1). First, the binding of the nanobodies was analyzed by using surface plasmon resonance (SPR) spectroscopy and binding affinities were determined by applying multi-cycle kinetics (Fig. 1c). Nanobodies VHH$_{GSDMD-1, -2, -3,}$ and $_{-5}$ bound to GSDMD with high affinities in the nanomolar range and displayed rapid association and slow dissociation rates. The tightest binder was VHH$_{GSDMD-1}$ with a dissociation constant ($K_D$) derived from the association and dissociation rate constants of 0.31 nM. VHH$_{GSDMD-2, -3}$ and $_{-5}$ had $K_D$s of 0.64 nM, 0.55 nM, and 0.49 nM, respectively. In contrast, VHH$_{GSDMD-6}$ and VHH$_{GSDMD-4}$ exhibited significantly lower

binding affinities in the medium to high nanomolar range. The association and dissociation rate constants and the $K_D$s derived from the SPR measurements are listed in Supplementary Table 1. Due to its low affinity, VHH$_{GSDMD-4}$ was excluded from further SPR experiments.

Binding epitopes of the nanobodies on GSDMD were analyzed using an SPR-based epitope binning assay (Fig. 1d–f and Supplementary Fig. 2). Chemically biotinylated GSDMD was immobilized on an SPR sensor chip and nanobodies were injected as analytes in a pair-wise manner to investigate the possibility of mutually exclusive binding or simultaneous binding of both nanobodies to GSDMD (Fig. 1d). VHH$_{GSDMD-1}$ and VHH$_{GSDMD-5}$ exhibited mutually exclusive binding with all other nanobodies (Supplementary Fig. 2). VHH$_{GSDMD-2}$ and VHH$_{GSDMD-3}$ bound mutually exclusive, but for both nanobodies, additional binding of VHH$_{GSDMD-6}$ was observed (Fig. 1e). According to these observations, VHH$_{GSDMD-1}$ and VHH$_{GSDMD-5}$, as well as VHH$_{GSDMD-2}$ and VHH$_{GSDMD-3}$, were grouped into one epitope bin, whereas VHH$_{GSDMD-6}$ stands alone (Fig. 1f).

### Two nanobodies inhibit the assembly of functional GSDMD pores in vitro

We used a liposome leakage assay to test whether nanobody binding affects the formation of functional GSDMD pores. GSDMD and nanobodies were added in equimolar ratios to calcein-packed liposomes and after the addition of caspase-4, calcein release through GSDMD pores was followed by measuring the fluorescence at 525 nm (Fig. 2a). As a control, we used the caspase inhibitor VX-765 which inhibited the calcein release completely (Fig. 2b). The addition of VHH$_{GSDMD-1}$ inhibited the calcein release to the same extent as the addition of VX-765, indicating that the assembly of functional GSDMD pores was nearly completely abrogated. VHH$_{GSDMD-2}$ also had an inhibitory effect, although to a lesser extent than VHH$_{GSDMD-1}$, and reduced the dye leakage by about 35% after three hours incubation at 37 °C compared to the sample without nanobody addition. The addition of VHH$_{GSDMD-6}$ had a small inhibitory effect and reduced the maximal fluorescence by 15% after the same time period. VHH$_{GSDMD-3, -4}$ and $_{-5}$ rather tended to increase the calcein leakage and had no inhibitory effect on GSDMD pore formation (Fig. 2b).

The degree of GSDMD inhibition at high concentrations of nanobody was determined by adding the nanobodies at a concentration of 10 μM (20:1 ratio to GSDMD) to the leakage assay (Fig. 2c). At this high concentration, VHH$_{GSDMD-1}$, VHH$_{GSDMD-2}$, VHH$_{GSDMD-6}$ inhibited the GSDMD pore formation by 87%, 75% and 40%, respectively. In contrast VHH$_{GSDMD-3}$, VHH$_{GSDMD-4}$ and VHH$_{GSDMD-5}$ yielded nearly no inhibition. Addition of the nanobodies to the assay alone without adding GSDMD or caspase-4 instead did not destabilize the liposomes significantly confirming that the fluorescence leakage is mediated by GSDMD and caspase-4 (Fig. 2d). The IC$_{50}$ values for the inhibitors VHH$_{GSDMD-1}$, VHH$_{GSDMD-2}$ and VHH$_{GSDMD-6}$ were obtained by adding the nanobodies in a dose-response measurement from 10 nM to 10 μM to 0.5 μM full length GSDMD. For VHH$_{GSDMD-1}$ an IC$_{50}$ value of $0.22 \pm 0.01$ μM was determined, while the value obtained for VHH$_{GSDMD-2}$ was $0.65 \pm 0.11$ μM and for VHH$_{GSDMD-6}$ $1.30 \pm 0.52$ μM (Fig. 2e–g).

We further analyzed the thermal stability of the nanobodies and their impact on the thermostability of GSDMD using a thermal shift assay by nano-differential scanning fluorimetry (nanoDSF). The nanobodies were titrated to GSDMD in increasing concentrations, revealing a peak of fluorescence shift distinct from the peaks observed for GSDMD or nanobody alone, indicating complex formation (Supplementary Fig. 3). At equimolar concentrations, the two strongly inhibiting nanobodies (VHH$_{GSDMD-1}$, VHH$_{GSDMD-2}$) increased the thermal stability of GSDMD by up to 9.4 °C (Fig. 2h). In contrast, the non-inhibitory nanobodies VHH$_{GSDMD-4}$ and VHH$_{GSDMD-5}$ had a slightly destabilizing effect and decreased the melting temperature of GSDMD by up to 2.5 °C. VHH$_{GSDMD-3}$ and VHH$_{GSDMD-6}$ increased the thermal stability of GSDMD by 4.2 °C.

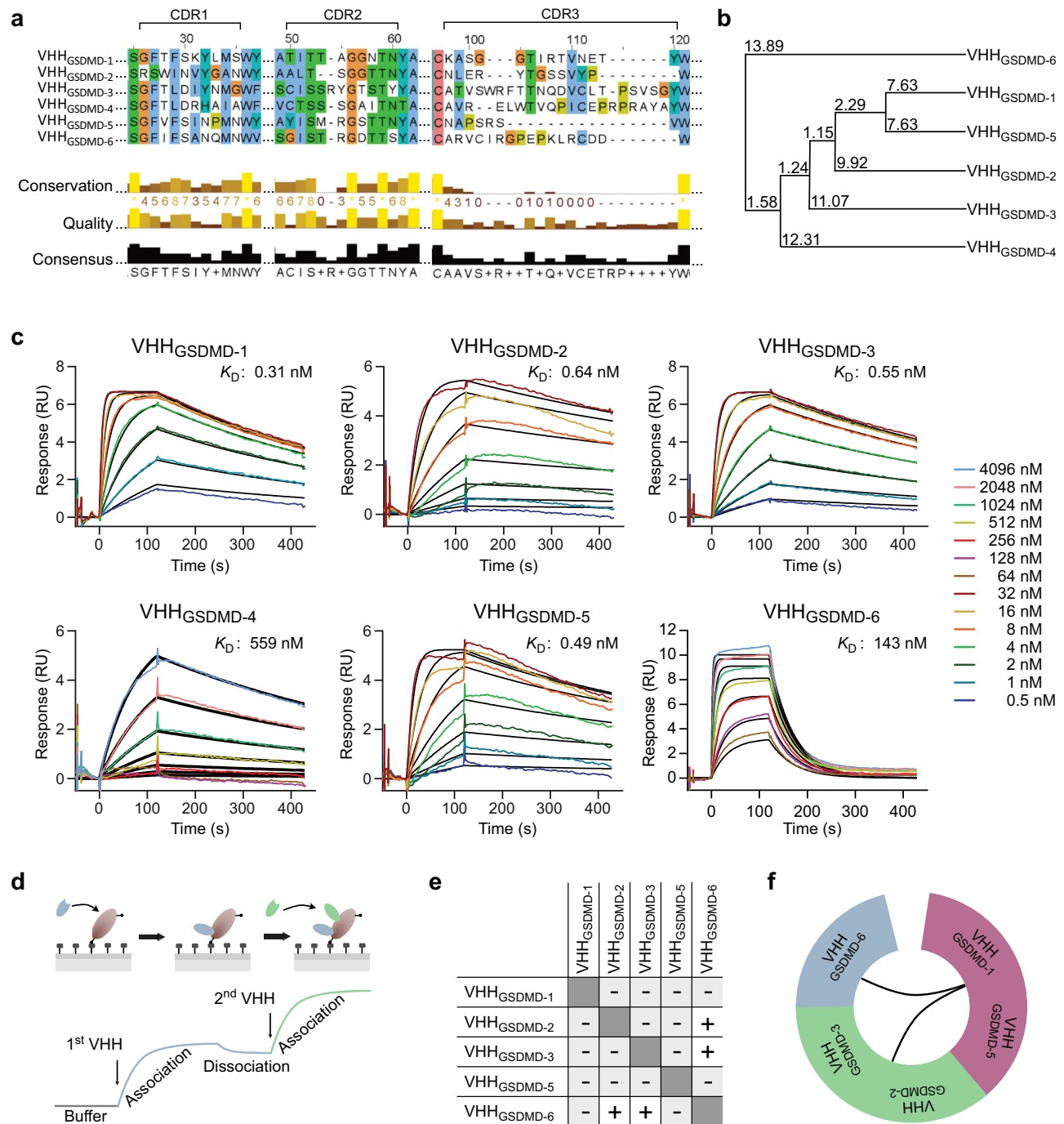

**Fig. 1 | Identification of six human GSDMD targeting nanobodies. a** Amino acid sequence alignment of the GSDMD targeting nanobodies showing the three complementarity determining regions (CDR1-3). **b** Average distance tree based on the amino acid sequence of the nanobodies. The tree displays the average distance using percent identity and was calculated using the software Jalview. **c** Determination of binding affinities using surface plasmon resonance (SPR). Chemically biotinylated GSDMD was immobilized on a sensor chip and nanobodies were injected as analytes at the indicated concentrations for 120 s, followed by dissociation for 300 s. Dissociation constants ($K_D$s) were determined from the association and dissociation fits by applying a 1:1 binding model. Source data are provided as a Source Data file. **d** Epitope binning assay. Chemically biotinylated GSDMD was immobilized on an SPR sensor chip and the competitive binding of nanobodies was tested in a pairwise manner. Association of the second nanobody to a distinct epitope can be observed as a second association event in the SPR sensorgram. **e** Interaction matrix of VHH$_{GSDMD-1}$ to -6. **f** Binning of the nanobodies according to their properties in the competitive binding assay. Nanobodies VHH$_{GSDMD-1}$ and -5 are grouped into one bin, as they showed mutually exclusive binding with all other nanobodies. VHH$_{GSDMD-2}$ and -3 showed mutually exclusive binding to one another but allowed simultaneous binding of VHH$_{GSDMD-6}$, which itself does not share any binding similarities with the other nanobodies.

## Crystal structure of GSDMD in complex with two nanobodies

To map the epitopes of the nanobodies in detail and to shed light on the molecular mechanism by which VHH$_{GSDMD-1}$ and $_{-2}$ inhibit GSDMD pore formation, we initiated crystallization studies of the GSDMD-nanobody complexes. For this, we used a GSDMD construct where the linker region between the two domains (residues 247–272) and residues 184-194 in the NTD were deleted to prevent precipitation during crystallization following a previous report[10]. Crystallization trials were successful for a tripartite complex consisting of GSDMD, VHH$_{GSDMD-2}$ and VHH$_{GSDMD-6}$, and well-diffracting crystals were reproducibly grown

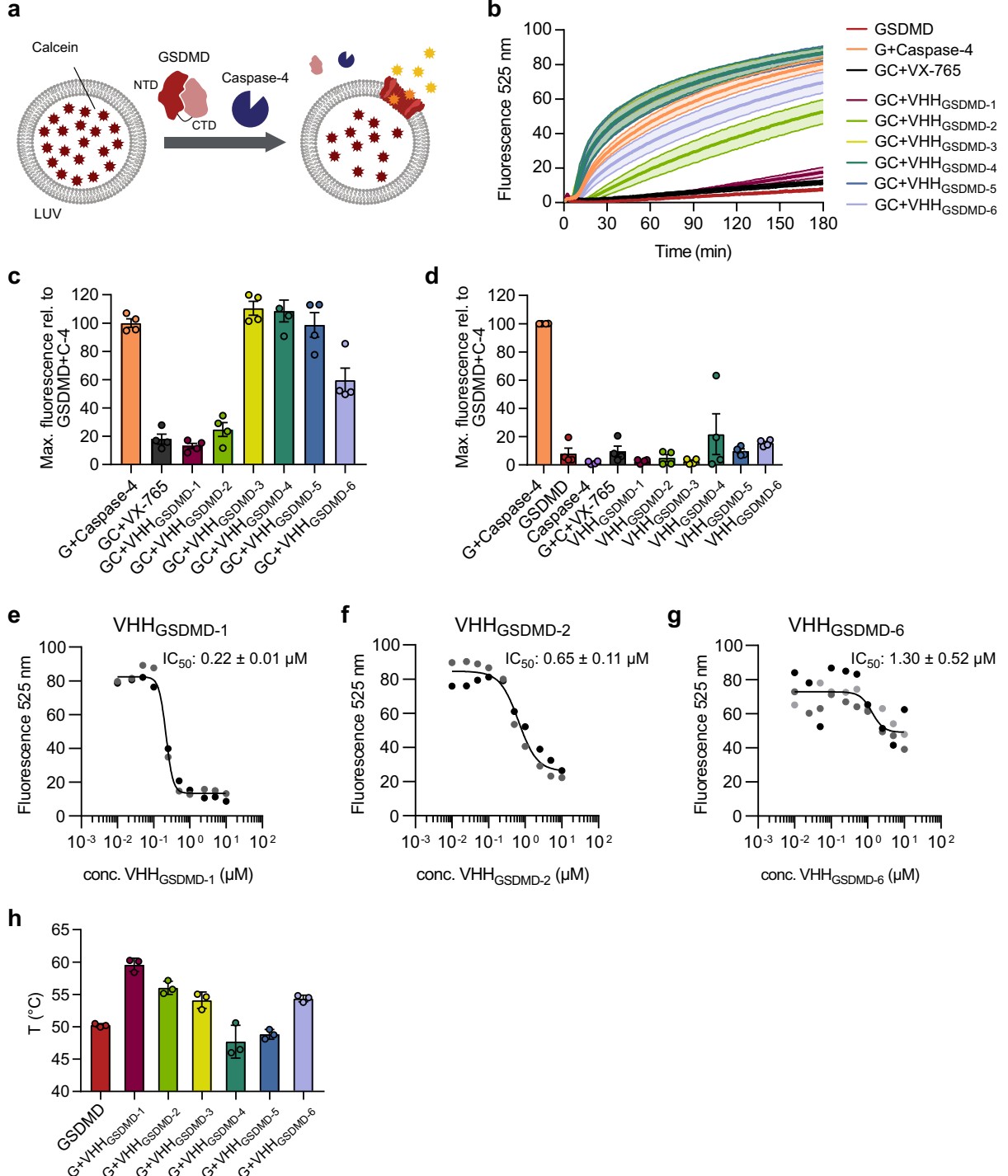

**Fig. 2 | VHH_GSDMD-1 and VHH_GSDMD-2 inhibit the formation of functional GSDMD pores in vitro. a** Liposomes composed of POPC, PE, and CL in a 32:55:13 ratio were loaded with the self-quenching dye calcein. GSDMD and nanobodies were added in equimolar ratios (0.5 μM). After the addition of 0.2 μM caspase-4, calcein release was observed by detecting the fluorescence emitted at 525 nm after excitation at 485 nm. **b** Liposomes, GSDMD (G), nanobodies, and caspase-4 (C) were incubated at 37 °C for 180 min and calcein release was detected every minute. VX-765 was used at 0.125 μM concentration. $N = 5$ independent experiments, the mean ± SEM is shown. **c** The nanobodies were added in a 20:1 ratio (10 μM) to GSDMD to the leakage assay. The maximal fluorescence after 180 min is shown relative to the maximal fluorescence obtained for the GSDMD+Caspase-4 sample. $N = 4$ independent experiments, the mean ± SEM is shown. **d** The effect of the nanobodies alone in the fluorescence leakage assay was testes as control. The nanobodies were added

to the assay at a concentration of 10 μM without adding GSDMD or caspase-4. $N = 4$ independent experiments, the mean ± SEM is shown. **e** Dose response curves of VHH_GSDMD-1 in the liposome leakage assay. $N = 2$ independent experiments, indicated by two different shades of grey for the data points. **f** Dose response curves of VHH_GSDMD-2 in the liposome leakage assay. $N = 2$ independent experiments, indicated by two different shades of grey for the data points. **g** Dose response curves of VHH_GSDMD-6 in the liposome leakage assay. $N = 3$ independent experiments, indicated by three different shades of grey for the data points. **h** Melting temperatures of GSDMD, the nanobodies and GSDMD–nanobody complexes were determined using nanoDSF. The unfolding temperatures of GSDMD at 5 μM and the GSDMD nanobody complexes after addition of 5 μM nanobody are shown. $N = 3$ independent experiments, data represented as mean ± SD. Source data are provided as a Source Data file.

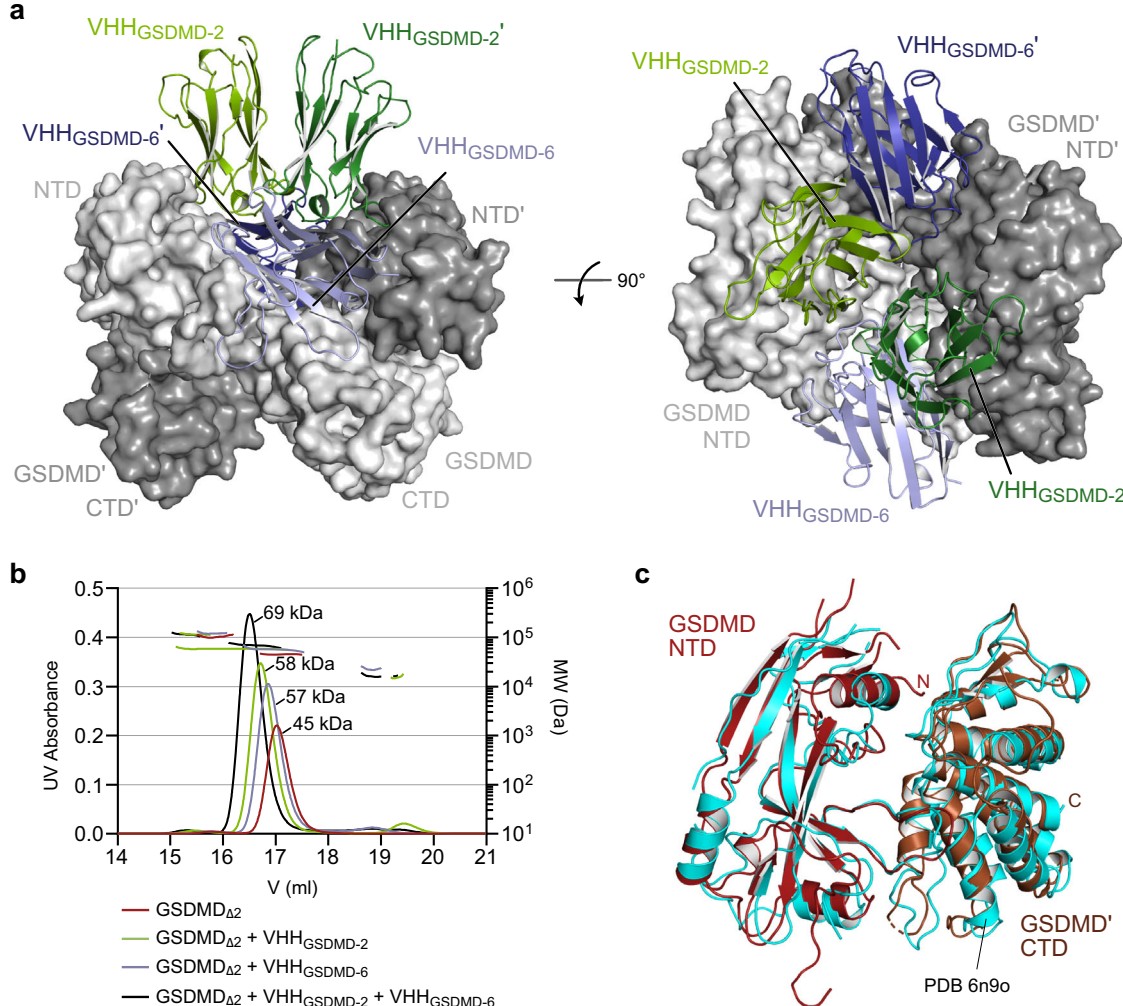

**Fig. 3 | Crystal structure of GSDMD in complex with VHH$_{GSDMD-2}$ and VHH$_{GSDMD-6}$. a** Structure of the human GSDMD–VHH$_{GSDMD-2}$–VHH$_{GSDMD-6}$ complex consisting of two heterotrimeric complexes. The two GSDMD proteins are shown in surface representation (light and middle grey) with nanobodies VHH$_{GSDMD-2}$ (green) and VHH$_{GSDMD-6}$ (blue) shown as ribbon diagram. **b** SEC-MALS analysis of the GSDMD–VHH$_{GSDMD-2}$–VHH$_{GSDMD-6}$ complex using a Superose 6 GL 10/300 column confirms the heterotrimeric assembly in solution. Source data are provided as a Source Data file. **c** Superimposition of the complex of GSDMD NTD and GSDMD' CTD determined here (PDB 7z1x, brown) with the previous GSDMD crystal structure (PDB 6n9o, cyan).

using this combination of proteins. We determined the crystal structure of the complex at 1.9 Å resolution by molecular replacement using the structures of human GSDMD (PDB 6n9o)[10] and a BC2 nanobody (PDB 5ivo)[38] as search models. GSDMD and nanobodies are found in 1:1:1 stoichiometry with the two nanobodies unambiguously identified by their characteristic CDR regions. Two heterotrimeric GSDMD–VHH$_{GSDMD-2}$–VHH$_{GSDMD-6}$ complexes form the asymmetric unit of the crystal lattice and were refined to a R$_{work}$ of 21.2% and R$_{free}$ of 24.9% with good stereochemistry (Fig. 3a and Supplementary Table 2). The two GSDMD molecules found in the structure form a dimeric complex in which the NTD of one GSDMD molecule is tightly interacting with the CTD of the other, resulting in a buried surface area of 4018 Å² counting both molecules. Looking at a single heterotrimeric complex, VHH$_{GSDMD-2}$ is bound to the NTD of GSDMD, whereas VHH$_{GSDMD-6}$ interacts with the NTD and the CTD as well as the connecting linker region between both domains, stabilizing the twinned assembly of the mixed NTD–CTD' and NTD'–CTD formation (Supplementary Fig. 4a).

Since GSDMD has not been reported to form dimers before, we hypothesized that the complex formation observed might be a crystallographic artifact. To substantiate our hypothesis, we performed size exclusion chromatography coupled to multi-angle light

scattering (SEC-MALS) analysis. The complex of GSDMD, VHH$_{GSDMD-2}$, and VHH$_{GSDMD-6}$ displayed an apparent molecular weight of 69.2 kDa, consistent with a 1:1:1 complex with a calculated molecular weight of 79.7 kDa (Fig. 3b). For this reason, we conclude that the dimerization of the two heterotrimeric complexes occurred during crystallization. The interaction of the mutually twisted N- and C-terminal domains of the two GSDMD molecules resembles the interactions between both domains observed in the previously determined crystal structure of human GSDMD (PDB 6n9o)[10]. Superimposition of our NTD–CTD' complex with the previous structure results in a root mean square deviation (RMSD) of 2.17 Å over 333 atoms, while the NTD alone overlays with an RMSD value of 1.14 Å over 136 atoms and the CTD with 1.15 Å over 186 atoms indicating excellent conformity (Fig. 3c).

**Binding interfaces of GSDMD–VHH interactions**
The CDR1, -2 and -3 segments of VHH$_{GSDMD-2}$ comprise 10, 9 and 12 residues, respectively, and the nanobody backbone is stabilized by a conserved disulfide bond between C22 and C95. The interface of VHH$_{GSDMD-2}$ and GSDMD is mainly built by CDR1 and CDR3 of VHH$_{GSDMD-2}$, whereas CDR2 does not contribute significantly to the interaction. In contrast, for VHH$_{GSDMD-6}$ all three CDRs are involved in

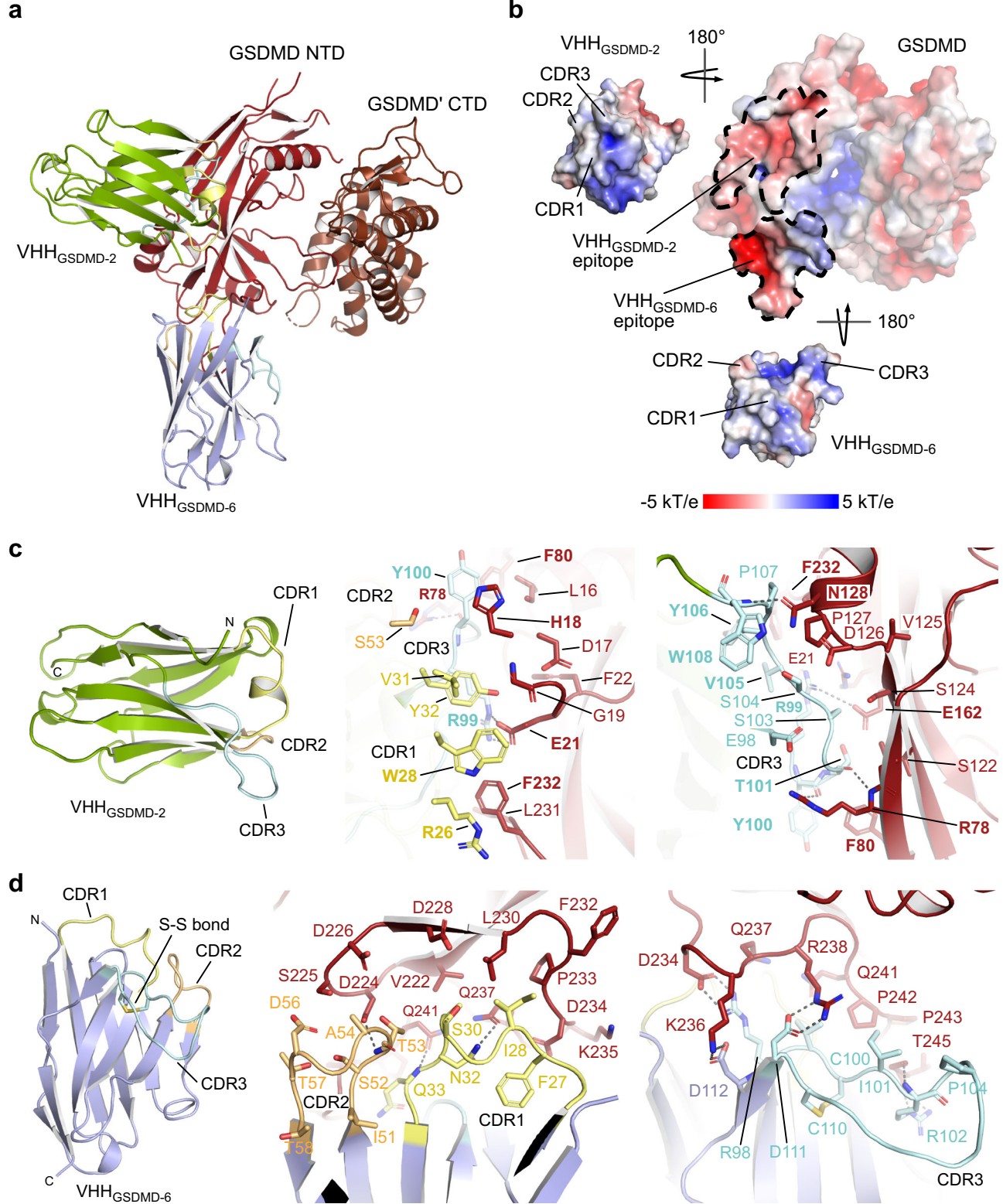

**Fig. 4 | Binding interfaces between GSDMD and VHH$_{GSDMD-2}$ and VHH$_{GSDMD-6}$.**
**a** Cartoon representation of the GSDMD NTD/GSDMD' CTD–VHH$_{GSDMD-2}$–VHH$_{GSDMD-6}$ structure showing GSDMD, GSDMD' NTD, VHH$_{GSDMD-2}$ and VHH$_{GSDMD-6}$. The CDR1, -2, and -3 regions of the nanobodies are highlighted in yellow, orange, and cyan, respectively. **b** Electrostatic surface potential of the GSDMD−nanobody complex. The VHH$_{GSDMD-2}$ and VHH$_{GSDMD-6}$ binding epitopes on the surface of GSDMD are marked with dotted lines. The complementary interfaces are displayed in the "open book" conformation. **c** Details of the VHH$_{GSDMD-2}$ to GSDMD interface. The CDR1 (yellow) and CDR2 (orange) interactions with the NTD are shown in the middle panel, while the CRD3 (cyan) interactions are shown in the right panel. **d** Details of the VHH$_{GSDMD-6}$ to GSDMD interface. The CDR interactions with the NTD and linker region of GSDMD are shown in the middle and right panel following the same color coding as in c.

binding to GSDMD (Fig. 4a). Electrostatic interactions are crucial for the interaction of both nanobodies with GSDMD. The positively charged CDRs of VHH$_{GSDMD-2}$ bind to an acidic cleft on the GSDMD surface involving residues E15, D17, E21, D126, and E162. The CDRs of VHH$_{GSDMD-6}$ contact with the acidic residues D224, D226, D228, D234, and D275, and the VHH$_{GSDMD-6}$ backbone contacts residues E448 and E459 on the GSDMD surface (Fig. 4b).

Binding of VHH$_{GSDMD-2}$ to GSDMD results in a buried surface area of 1521 Å$^2$ counting both molecules. A pronounced salt bridge is formed between R99 in CDR3 of VHH$_{GSDMD-2}$ and E21 in the GSDMD NTD complemented at the opposite side by a weak interaction to E162 of GSDMD (Fig. 4c). Moreover, the backbone carboxyl groups of neighboring residues Y100 and T101 in the CDR3 form intermolecular hydrogen bonds with R78 of GSDMD. Another hydrogen bond is formed between the CDR3 residue W108 and N128 on the GSDMD surface. Supporting hydrophobic contacts are formed between F232 of GSDMD with V105, Y106, R26 and W28 of VHH$_{GSDMD-2}$, as well as Y100 in the CDR3 of VHH$_{GSDMD-2}$, which is sandwiched between H18 and F80 of GSDMD.

VHH$_{GSDMD-6}$ was indispensable for the crystallization of high-resolution GSDMD-nanobody complexes by serving as a crystallization chaperone. The buried surface area of this interaction is exceptionally large with 2615 Å$^2$ (counting both molecules), with the CDRs of VHH$_{GSDMD-6}$ targeting the NTD of GSDMD, while the β-barrel side of the IgV fold interacts with the CTD. This two-sided interaction is only possible by the mixed assembly of the N- and C-terminal domains of GSDMD in the formation of two GSDMD–VHH$_{GSDMD-2}$–VHH$_{GSDMD-6}$ complexes in the asymmetric unit and might be the reason for the twist of the two subdomains (Supplementary Fig. 4a). The CDR3 of VHH$_{GSDMD-6}$ is particularly long comprising 15 residues and is stabilized by an additional disulfide bond between C100 and C110; a feature that helps to clearly distinguish the two nanobodies in the crystallographic electron density map (Supplementary Fig. 4b). The charged cluster D234, K236 and R238 towards the end of the NTD in GSDMD is targeted by the CDR3 through a tight salt bridge interaction with D111 (to R238), followed by R98 (to D234), and complemented with D112 (to K236) (Fig. 4d). R109 of the VHH$_{GSDMD-6}$ CDR3 instead loops to the CTD of GSDMD and interacts with E417 and the main chain carboxy group of Q411 (Supplementary Fig. 4c). Various additional interactions can be found between CDRs 1 and 2 and the NTD of GSDMD. Residues N32 and Q33 in the CDR1 form hydrogen bonds with Q237 and Q241, whereas CDR2 residue T53 contacts D224 on the GSDMD NTD (Fig. 4d). I101 forms hydrophobic interactions with the linker region between both GSDMD domains, while L45 in the loop opposing CDR1 and CDR2 of VHH$_{GSDMD-6}$ interacts with the CTD of GSDMD. Several other residues in the VHH$_{GSDMD-6}$ backbone comprising residues 39, 42–45, 47, 95, and 112–115 as well as residues 104–112 in the CDR3 contact the CTD of GSDMD which might contribute to the role of VHH$_{GSDMD-6}$ in facilitating crystallization (Supplementary Fig. 4d).

The interaction of VHH$_{GSDMD-1}$, VHH$_{GSDMD-2}$ and VHH$_{GSDMD-6}$ with endogenous GSDMD was tested in a pull-down assay using immobilized nanobodies and THP-1 cell lysate. All three nanobodies effectively interacted with endogenous GSDMD as observed by SDS PAGE analysis followed by zinc staining and western blotting (Supplementary Fig. 5). Potential cross-reactivity of the nanobodies with gasdermins B and E was tested using the gasdermins expressed as renilla-fusion proteins in HEK293T cells and the recombinant nanobodies fused to a C-terminal His-tag. By using an antibody directed against the His-tag of the nanobodies and renilla luciferase activity as readout for an effective pull-down, we found that VHH$_{GSDMD-1}$, VHH$_{GSDMD-2}$ and VHH$_{GSDMD-6}$ specifically interacted with GSDMD but not with GSDMB or GSDME. In addition, we analyzed the binding sites of the nanobodies VHH$_{GSDMD-2}$ and VHH$_{GSDMD-6}$ in a structure-based sequence alignment of all six human gasdermins (Supplementary Fig. 6). Indeed, the sequence variability in all gasdermins is very high and no conservation or homology in the binding epitopes of GSDMD to the nanobodies is observed for any other gasdermin. This observation agrees with the high specificity of the nanobodies, which target only human GSDMD.

## GSDMD pore formation is inhibited by blocking oligomerization of the GSDMD NTD

We have shown that VHH$_{GSDMD-1}$ and VHH$_{GSDMD-2}$ strongly inhibited the assembly of functional GSDMD pores in vitro, leading to the question, by which mechanism pore formation is abrogated. Both nanobodies were found to bind to an overlapping epitope on the GSDMD NTD in the SPR-based epitope binning experiment. We set up an in vitro caspase cleavage assay using recombinant full length GSDMD and human caspase-4. A time course experiment shows the decrease of full length GSDMD over time and the corresponding appearance of cleaved N- and C-terminal domains (Fig. 5a, left panel). Addition of the VHHs at a 1:1 molar ratio (VHH to GSDMD) revealed that both nanobodies did not affect GSDMD cleavage by caspase-4 as observed on SDS-PAGE (Fig. 5a, middle and right panels), suggesting that the mechanism of pyroptosis inhibition is not achieved by the inhibition of the cleavage reaction.

The cryo-EM structure of the human GSDMD pore was recently determined[13]. We superimposed our structure of the nanobody bound GSDMD NTD with the structure of one GSDMD subunit (PDB 6vfe) in the pore conformation (Fig. 5b, c). The superimposition shows that VHH$_{GSDMD-2}$ and VHH$_{GSDMD-6}$ bind to the globular part of the activated NTD. The basic batch in the NTD–CTD interface, required for membrane association after caspase cleavage, is fully exposed in the VHH-bound structure enabling the electrostatic interaction to negatively charged lipid surfaces (Fig. 5b). Whereas VHH$_{GSDMD-6}$ binds on top of the globular rim of the GSDMD pore and does interfere only weakly with oligomerization, VHH$_{GSDMD-2}$ binds in the oligomerization interface of the single N-termini and therefore sterically inhibits pore assembly (Fig. 5d). Therefore, we conclude that VHH$_{GSDMD-2}$ directly interferes with the oligomerization of the activated GSDMD NTD. VHH$_{GSDMD-1}$ shared an overlapping epitope with both VHH$_{GSDMD-2}$ and VHH$_{GSDMD-6}$ in the SPR-based epitope binning which suggests that also this nanobody binds to the globular rim of the GSDMD pore and likely also inhibits oligomerization.

## Discussion

GSDMD is the enforcer of pyroptosis, mediating the final common step of all inflammasome pathways. As pyroptosis is implicated in many diseases, a deep understanding of the mechanisms underlying GSDMD pore formation and its regulatory processes is essential. Inhibiting GSDMD is an attractive strategy to treat excessive inflammation and requires GSDMD-specific interacting molecules. To date, three small molecule inhibitors (necrosulfonamide, disulfiram and dimethyl fumarate) have been reported that covalently modify C191 in the GSDMD NTD and effectively prevent pyroptosis in cells and suppress inflammatory responses in murine models[30–32]. However, this class of inhibitors has a serious disadvantage: due to their cysteine reactivity, the compounds are not specific to GSDMD and binding to off-targets might cause unwanted side-effects in the human body[39–42]. In this study, we characterized six unique GSDMD targeting nanobodies that bind with varying affinities to the NTD of GSDMD. Two of these nanobodies (VHH$_{GSDMD-1}$ and $_{-2}$) inhibited GSDMD pore formation in an in vitro liposome leakage assay whereas one nanobody (VHH$_{GSDMD-6}$) weakly inhibited at high molar excess. The high-resolution crystal structure of GSDMD in complex with one inhibitory and the weakly inhibiting nanobody revealed that the inhibitory nanobody sterically blocks GSDMD pore assembly by binding to an epitope residing in the oligomerization interface of the activated GSDMD NTD. Whereas the side-to-side assembly of individual NTDs required for pore formation is inhibited by the nanobody VHH$_{GSDMD-2}$, caspase cleavage is not

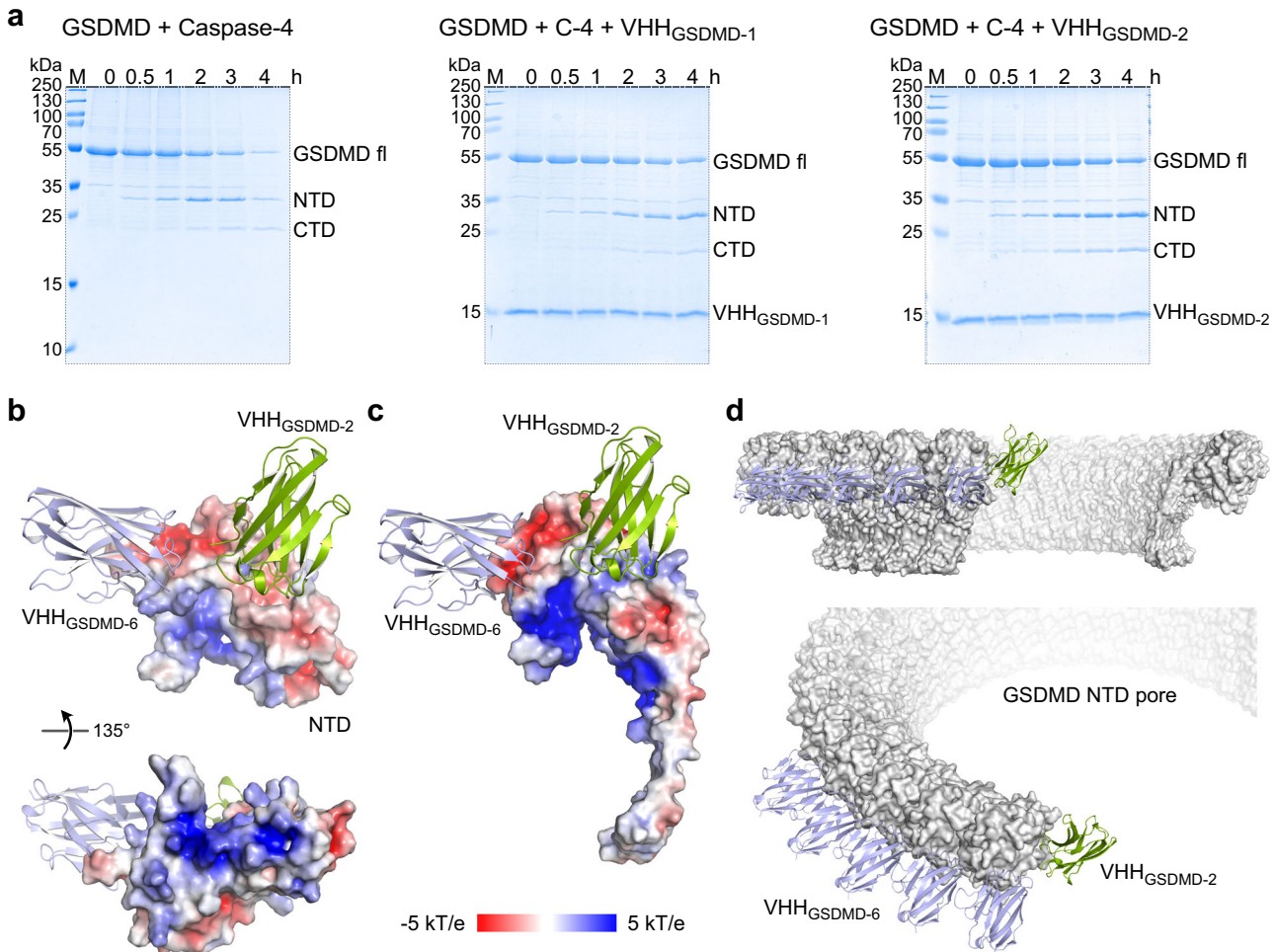

**Fig. 5 | Mechanism of pyroptosis inhibition by GSDMD targeting nanobodies.**
**a** Caspase cleavage of GSDMD is not inhibited by VHH$_{GSDMD-1}$ or $_{-2}$. Recombinant GSDMD was incubated with caspase-4 at 37 °C for 4 h either without or with an equimolar amount of VHH$_{GSDMD-1}$ or VHH$_{GSDMD-2}$. GSDMD cleavage into by caspase-4 into NTD and CTD was monitored by Coomassie-stained SDS-PAGE analysis. Representative gels of $N = 2$ experiments are shown. Source data are provided as a Source Data file. **b** The basic surface patch of the GSDMD NTD is not masked by VHH$_{GSDMD-2}$ and VHH$_{GSDMD-6}$ binding. **c** Superimposition of VHH$_{GSDMD-2}$ and $_{-6}$ nanobody binding to a single subunit of the pore forming GSDMD NTD (6vfe)[13] shown as electrostatic surface display. **d** Overlay of VHH binding to the cryo-EM structure of the GSDMD pore (6vfe)[13]. Whereas VHH$_{GSDMD-6}$ (lightblue) binds to an epitope of GSDMD NTD that locates at the outer rim of the pore, VHH$_{GSDMD-2}$ (green) binds to the oligomerization interface in the pore assembly of the NTD subunits.

affected and the basic patches required for membrane binding remain fully exposed.

By sterically inhibiting oligomerization instead of targeting reactive cysteine residues in GSDMD, our nanobodies provide an alternative mechanism of pyroptosis inhibition. GSDMD is an intracellular protein and the delivery of antibodies and nanobodies to the cytoplasm has long been limited due to their inability to cross the plasma membrane[43,44]. Transient GSDMD pore formation after inflammasome activation seems to be able to allow access of inhibitory GSDMD nanobodies to the cytosol and prevents cell death by pyroptosis[37]. In addition, recent studies showed that nanobodies can be delivered to the cytoplasm of cells using cell penetrating peptide fusions, and nanobody mRNA could be delivered using gene therapy approaches[44–48]. Our SPR-based binding experiments were performed in the presence of 5 mM DTT, indicating that the nanobodies retain their binding ability even under reducing conditions and should not be affected by the reducing milieu of the cytoplasm. This observation is consistent with the inhibition of GSDMD pore formation in THP-1 cells.

Gasdermins share a low sequence similarity ranging from 23.9 to 49.4% within the human gasdermin protein family[2]. This diversity includes the known cleavage sites in human and mouse gasdermins that are targeted by various proteases[49]. Given the large binding epitopes of the nanobodies on the target proteins that typically comprise more than 18 residues and involve hydrogen bond and salt bridge formations, it is not surprising that the nanobodies identified here target specifically only human GSDMD but not any other gasdermin (Supplementary Figs 5, 6). As a side note, three homologous changes in the binding epitope from mouse to human NINJ1 already prevented its recognition by a monoclonal antibody and abrogated the function as an inhibitor of membrane rupture[20]. The large buried surface area of usually more than 1,500 Å$^2$ and the high binding affinity of the interaction with dissociation constants in the low nanomolar range therefore cause a specific interaction of the nanobody with the target protein.

Rather than being applied as drugs themselves, our nanobodies could also facilitate the development and characterization of GSDMD-specific small molecule inhibitors. Due to their function as crystallization chaperones, allowing the generation of fast growing, reproducible and well diffracting GSDMD crystals for X-ray crystallography and determination of high-resolution structures, they could act as versatile tools. In fact, we have re-engineered the first loop section 184-194 back into the GSDMD crystallization construct and reproducibly grown crystals that diffract up to 2.1 Å resolution having the same space group and unit cell parameters, albeit without showing

electron density for this section. Apart from that, the structure resembles the one described here and consists of a twisted dimer of a trimeric GSDMD–VHH$_{GSDMD-2}$–VHH$_{GSDMD-6}$ complex. Moreover, the nanobodies may help to stabilize GSDMD NTD monomers, which could be used in compound identification screenings, e.g., from DNA-encoded chemical library screens. Interestingly, VHH$_{GSDMD-2}$, but not VHH$_{GSDMD-6}$, binds to a similar surface patch on the GSDMD NTD as the *Shigella flexneri* ubiquitin ligase IpaH7.8, suggesting mutually exclusive binding modes. IpaH7.8, which targets GSDMD for degradation to prevent pyroptosis and enable infection[50], has recently been shown to bind to the NTD of both GSDMB and GSDMD through its LRR domain[51,52]. However, as the nanobody binds to GSDMD with a significantly higher affinity than IpaH7.8, the displacement of the ubiquitin ligase should prevent its degradation, underlining its application as a biological tool, for example for the characterization of bacterial toxins. We here describe the identification of inhibitory and non-inhibitory nanobodies targeting human GSDMD that can be used in a variety of ways from studying GSDMD biology and pyroptosis to compound screening approaches or the application as anti-inflammasomal inhibitors.

## Methods

### Protein expression and purification

The coding sequence for human wild-type, full length GSDMD 1-484 was cloned into a pET-28a-based bacterial expression vector providing an N-terminal His$_6$-SUMO-tag. The expression constructs were transformed in *E. coli* Rosetta (DE3) cells, and the cells were grown in 2x-LB-medium containing 0.5% glucose and 50 μg/ml kanamycin at 37 °C until the OD$_{600}$ reached 0.8. Expression was induced overnight at 20 °C with 0.2 mM isopropyl β-D-1-thiogalactopyranoside (IPTG) and 0.6% (w/v) lactose. Cells were harvested and lysed by sonication in a lysis buffer containing 25 mM Tris (pH 8.0), 200 mM NaCl, 10% glycerol, 10 mM imidazole, and 5 mM DTT. Cell lysates were cleared by centrifugation at 36,000 × g for 45 min and the His-SUMO fusion proteins were enriched on Ni-NTA beads (Thermo Fisher Scientific). The protein was eluted using a buffer containing 25 mM Tris (pH 8.0), 200 mM NaCl, 300 mM imidazole, and 5 mM DTT, and a buffer exchange to 25 mM Tris (pH 8.0), 200 mM NaCl, 5% glycerol and 5 mM DTT was performed using a PD10 column (Cytiva). The samples were incubated with the SUMO protease ULP1 (homemade) at 4°C overnight, followed by a second Ni-NTA chromatography to remove uncleaved protein, ULP1 protease, and the His-SUMO-tag. Subsequently, the samples were subjected to size-exclusion chromatography using an S200 column (GE Healthcare) and a buffer containing 20 mM HEPES (pH 8.0), 200 mM NaCl and 5 mM DTT. For crystallization experiments, residues 247–272 (GSDMD$_{Δ247-272}$) or residues 184–194 and 247–272 (GSDMD$_{Δ184-194/Δ247-272}$) were deleted to prevent precipitation during crystallization based on previous reports[7–10,53].

Nanobodies were expressed according to a routine protocol[54]. Briefly, *E. coli* WK6 cells were transformed with the pHEN6 vectors for bacterial, periplasmic expression of GSDMD-targeting nanobodies. VHH$_{GSDMD-1, -2}$ and $_{-3}$ were expressed with a C-terminal LPTEG-His-tag, and VHH$_{GSDMD-4, -5}$, and $_{-6}$ were expressed with a C-terminal HA-His-tag. Cells were grown at 37 °C in TB medium containing 100 μg/ml ampicillin until the OD$_{600}$ reached 0.6. Expression was induced overnight with 1 mM IPTG at 20 °C. Cells were harvested and periplasmic extracts were generated using osmotic shock. For this purpose, cell pellets were resuspended in TES buffer (20 mM Tris (pH 8.0), 0.65 mM EDTA, 0.5 M sucrose) and incubated for 1 h, followed by incubation in 0.25x TES for at least 1 h at 4 °C. Lysates were cleared by centrifugation at 36,000 x g for 45 min and the His-tagged nanobodies were enriched using Ni-NTA beads. Beads were washed using a buffer containing 50 mM Tris (pH 7.5, 150 mM NaCl, and 10 mM) imidazole and proteins were eluted in the same buffer supplemented with 0.5 M imidazole. Protein containing elution fractions were pooled and subjected to gel

filtration in a buffer containing 20 mM HEPES pH 7.5, and 150 mM NaCl using an S75 16/600 column (GE Healthcare).

The coding sequence for human caspase-4 was cloned into the pACEBac1-His-SUMO expression vector. Plasmids were amplified in *E. coli* DH10 cells. Baculoviruses were produced by transfection of the bacmid DNA into *Sf*9 insect cells, using the transfection reagent cellfectin (Mirus Bio, Madison WI). The transfection was carried out in a 6-well format with 0.7 × 10$^6$ cells/well and cells were incubated at 27 °C. After three days, the initial virus stock V$_0$ was harvested and used to infect a 20 ml culture of *Sf*9 cells grown to a density of 0.6 × 10$^6$ cells/ ml. V$_1$ viruses were harvested after three days and an *Sf*9 culture grown to a density of 1.0 × 10$^6$ cells/ml was infected with 2% V$_1$. After another three days, the V$_2$ viruses were harvested. Expression cultures were infected at a cell density of 1.5 × 10$^6$ cells/ml with 1% virus and proteins were expressed for 48 h at 27 °C. Cells were harvested and lysed by sonication in a buffer containing 50 mM Tris (pH 8.0), 150 mM NaCl, 5 mM imidazole, and 2 mM β-ME. Cell lysates were cleared by centrifugation at 36,000 × g for 45 min. His-SUMO-caspase-4 was purified using Ni-NTA affinity chromatography. Elution fractions containing the His-SUMO-fusion protein were pooled and concentrated in a 30 K Amicon to a concentration of 10 mg/ml. The sample was incubated at 4°C overnight to enhance the auto-activation of the protease.

### Protein crystallization and data collection

Screening of crystallization conditions for wild type GSDMD and GSDMD$_{Δ184-194/Δ247-272}$ in complex with nanobodies VHH$_{GSDMD-1}$ to VHH$_{GSDMD-6}$ and the combination of VHH$_{GSDMD-2}$ plus VHH$_{GSDMD-6}$ was performed using commercial kits from Molecular Dimensions (Maumee, OH, USA) and Jena Bioscience (Jena, Germany) with the sitting drop vapor diffusion method. Initial crystals of the sample containing GSDMD$_{Δ184-194/Δ247-272}$ in complex with VHH$_{GSDMD-2}$ and VHH$_{GSDMD-6}$ were obtained at a protein concentration of 20 mg/ml using a reservoir solution containing 0.07 M NaCl, 22% (v/v) PEG 400 and 0.05 M Na$_3$Cit pH 4.5 at 20 °C. Optimization of crystallization conditions led to well diffracting crystals grown at 20 mg/ml in a reservoir solution consisting of 0.04 M NaCl, 25.8% (v/v) PEG 400 and 0.05 M Na$_3$Cit pH 4.4 at 20 °C.

Crystals were frozen in the reservoir solutions plus PEG 400 at a final concentration of 35% in liquid nitrogen. X-ray diffraction data were collected at beamline P13 of the PETRA III synchrotron at "Deutsches Elektronen-Synchrotron" (DESY) in Hamburg, Germany, at a wavelength of λ = 0.976255 Å. Diffraction data were processed in space group P3$_1$ with the program XDS[55]. The phases were determined by molecular replacement. For GSDMD, the previous crystal structure of human GSDMD (PDB 6n9o) was used as a search model. To account for possible movements between the N- and C-terminal domains of GSDMD, the structure was split into the GSDMD-NTD or -CTD, resulting in two separate search models. In addition, the structure of the BC2 nanobody (PDB 5ivo) was used as a search model for VHH$_{GSDMD-2}$ and VHH$_{GSDMD-6}$. The crystals were twinned and the twin law: k, h, -l, was employed during refinement with Phenix. Manual model building and refinement were performed with Coot[56] and Phenix[57], respectively. The crystal structures were validated by the MolProbity[58] server. Structure figures were prepared using PyMOL (The PyMOL Molecular Graphics System, Version 2.0 Schrödinger, LLC). Interaction analysis of the binding interfaces was performed using PDBePISA[59]. Gasdermin sequences were aligned with MultAlin and the secondary structure annotated with ESPript[60].

### Multi-angle light scattering (MALS)

For SEC-MALS analysis of the GSDMD–VHH$_{GSDMD-2}$–VHH$_{GSDMD-6}$ complex formation, GSDMD$_{Δ184-194/Δ247-272}$ was mixed with VHH$_{GSDMD-2}$, VHH$_{GSDMD-6}$ or both nanobodies at equimolar concentrations (189 μM) and injected into a Superose 6 10/300 GL column equilibrated with GSDMD-SEC buffer. The chromatography system was attached to a

three-angle light scattering detector (miniDAWN, Wyatt) and a refractive index detector (Optilab T-rEX, Wyatt). Data were collected every 0.5 s with a flow rate of 0.5 ml/min and analysed using the ASTRA V software (Wyatt).

## Surface plasmon resonance (SPR)

Surface plasmon resonance experiments were performed using a Biacore 8 K instrument (GE Healthcare). The flow system was cleaned using the maintenance "Desorb" function (Desorb Kit, GE Healthcare). The system was flushed with running buffer (20 mM HEPES (pH 8.0), 200 mM NaCl, 5 mM DTT, 0.05% Tween20) and all steps were performed at 25 °C chip temperature. Chemically biotinylated GSDMD at a concentration of 100 nM was immobilized for 180 s on flow cell 2 of a Series S Sensor Chip CAP using a biotin capture kit and a flow rate of 30 μl/min. The system was washed for 600 s with running buffer with a flow rate of 30 μl/min. Binding affinities were determined using multicycle kinetics. To account for different binding affinities, the nanobodies were injected at various concentrations (VHH$_{GSDMD-1, -2, -3, -5}$: 0.5–32 nM, VHH$_{GSDMD-4}$, and VHH$_{GSDMD-6}$: 64–4096 nM) with a flow rate of 30 μl/min. The association step was carried out for 120 s and the dissociation step for 300 s.

For biotinylation, full length human GSDMD protein was transferred into modification buffer (100 mM NaH$_2$PO$_4$, 150 mM NaCl, pH 8.0) using Zeba™ Spin Desalting columns in accordance with the manufacturer's instructions. Biotinylation was performed using the ChromaLink™ Biotin Labeling kit from Solulink. Five equivalents of ChromaLink Biotin, dissolved in dimethylformamide (5 mg/ml), were added to the protein and incubated at room temperature for 90 minutes as per the manufacturer's recommendations. Any unreacted biotin was removed using Zeba™ Spin Desalting columns (Thermo-Fisher Scientific) that were pre-equilibrated with PBS. The protein in PBS was recovered, and the degree of labeling was assessed using a NanoDrop spectrophotometer (ThermoFisher Scientific) by measuring the absorbance at 280 nm and 354 nm and employing the E1% ChromaLink Biotin molar substitution Calculator. For the SPR experiments, GSDMD protein labeled with approximately 2 biotins per molecule was utilized.

For epitope binning experiments, the nanobodies were tested pairwise for competitive binding. The first analyte was injected at a concentration of 128 nM for 120 s with a flow rate of 10 μl/min. This step was followed by a dissociation step for 60 s. Then, a mixture of the first and the second analyte (both 128 nM) was injected for 120 s with a flowrate of 10 μl/min, followed by a dissociation step of 30 s. After each cycle, surfaces were regenerated for 120 s using the regeneration solution of the capture kit with a flow rate of 10 μl/min. Data were referenced by blank cycle (no analyte injected) and subtraction of the reference flow cell (flow cell 1). Data were analyzed using the Biacore Insight Evaluation Software. Dissociation constants were determined based on fits applying a 1:1 binding model.

## Liposome leakage assay

The lipids for the generation of LUVs were obtained from Avanti polar lipids and dissolved in chloroform to a final concentration of 25 mg/ml. Liposomes were generated by mixing 80 μl phosphatidylcholine (POPC), 128 μl phosphatidylethanolamine (POPE), and 64 μl cardiolipin in a glass tube[12]. The chloroform was evaporated under a steady stream of nitrogen and lipids were rehydrated in 400 μl of an 80 mM calcein solution in H$_2$O (pH 7.0). The liposome suspension was vortexed extensively and subjected to five freeze and thaw cycles followed by extrusion through a 100 nm pore diameter polycarbonate membrane 31-times using the Avanti mini-extruder (Avanti Polar Lipids, Inc., Alabaster, AL). The extruded liposomes were passed through a PD10 column equilibrated with 20 mM HEPES (pH 7.4), 150 mM NaCl and 1 mM EDTA to remove excess calcein. For this purpose, 100 μl of liposomes were loaded onto the column and

200 μl elution fractions were collected. The homogeneity and quality of obtained liposomes were controlled using dynamic light scattering (DLS) and evaluating the packaging by measuring the fluorescence at 525 nm after lysis with 1% Triton-X 100, respectively. Fractions containing liposomes of good quality were pooled and diluted 1:10 in buffer.

For the liposome leakage assay, 50 μl of the liposome solution, 0.5 μM GSDMD, 0.5 μM His-SUMO-caspase-4, and 0.5 μM VHH were mixed in a final volume of 200 μl in a dark-well glass bottom plate and incubated at 37 °C for 180 minutes. As control, the caspase inhibitor VX-765 was used at a concentration of 0.125 μM. Every minute, the fluorescence emitted at 525 nm upon excitation at 485 nm was measured using a plate reader.

The degree of inhibition at high nanobody concentrations was determined for all nanobodies added to GSDMD at a concentration of 10 μM (20:1 ratio). The sample containing GSDMD and caspase-4 but no nanobody was used to determine the maximal fluorescence (100%). For the determination of IC$_{50}$ values, the nanobodies were applied at concentrations ranging from 0–10 μM, using the similar setup as before with caspase-4 as cleaving protease and fluorescence measurements after 180 min of incubation.

## Caspase cleavage assay

Recombinant GSDMD (15 μM) was incubated with an equimolar amount of VHH$_{GSDMD-1}$ or VHH$_{GSDMD-2}$ and caspase-4 (6 μM) at 37 °C for 4 h. GSDMD cleavage by caspase-4 was analysed by SDS-PAGE at the indicated time points.

## Thermal shift assay

NanoDSF was used to determine the effect of the GSDMD-targeting nanobodies on the thermal stability of GSDMD. Samples containing varying concentrations of protein were loaded into glass capillaries and applied to the nanoDSF device Prometheus NT.48 (Nanotemper). The samples were heated from 20 to 90°C with a slope of 1.5 °C/min and the unfolding of the proteins was observed by detecting shifts in the fluorescence at 330 and 350 nm. Data were analysed using the Nanotemper PR.ThermControl software.

## Cell lines

Human embryonic kidney (HEK) 293 T cells (ATCC Cat# CRL-3216, RRID: CVCL_0063), were cultivated in DMEM medium (Thermo Fisher Scientific) supplemented with 10% FBS, 100 U/ml penicillin and 100 μg/ml streptomycin; THP-1 cells (ATCC TIB-202) were cultured in RPMI medium (Thermo Fisher Scientific) supplemented with 10% FBS 100 U/ml penicillin and 100 μg/ml streptomycin.

## Antibodies

The following antibodies were used: rabbit anti-GSDMB (Cell Signaling Cat #76349, RRID:AB_ 2799883), 1:1000 dilution; rabbit polyclonal anti-GSDMD (Atlas Antibodies Cat# HPA044487, RRID:AB_2678957), 1:500 dilution; rabbit anti-DFNA5/GSDME clone EPR19859 (Abcam Cat# ab215191, RRID:AB_2737000), 1:1000 dilution; mouse anti−6x-His-Tag (Thermo Fisher Scientific Cat# MA1-135, RRID:AB_2536841), 1:1000 dilution; mouse anti-β-actin (Santa Cruz Biotechnology Cat #sc-47778, RRID:AB_626632), 1:200 dilution; mouse-IgGk BP-HRP (Santa Cruz Biotechnology Cat #sc−516102, RRID:AB_2687626,), 1:5,000 dilution; mouse anti-rabbit IgG-HRP (Santa Cruz Biotechnology Cat #sc-2357, RRID:AB_628497), 1:10,000 dilution.

## Western blot analysis

Western blot analysis was used to detect the presence of the proteins of interest in THP-1 cells and transfected HEK293T cells. Cells were lysed in lysis buffer (50 mM Tris pH 7.5, 500 mM NaCl, 1% NP-40, Roche cOmplete™ Mini protease Inhibitor Cocktail) and cell debris was pelleted by centrifugation of the lysates for 15 min at 14,000 g and 4 °C.

The supernatant was transferred to a fresh tube and supplemented with a final concentration of 1X SDS-PAGE sample buffer and boiled for 5 min at 95 °C. Proteins were separated by SDS-PAGE using 12% self-made SDS-PAGE gels. Separated proteins were transferred to nitro-cellulose membranes (Cytiva) by semi-dry transfer. Membranes were blocked in 5% non-fat dry milk (NFDM) solution in PBS for 1 h and probed with the following primary antibody dilutions: anti-GSDMB 1:1000, anti-GSDMC 1:1000, anti-GSDMD 1:500, anti-GSDME 1:2000, anti-β-actin 1:200 in PBS-T over night at 4 °C. Immunoblots were then probed with HRP-coupled secondary antibodies in PBS-T (1:5000) for 1 h at RT. Chemiluminescent signal was induced using Immobilon® Forte Western HRP Substrate (Sigma-Aldrich). Chemiluminescence was detected using the ChemiDoc imaging system (Bio-Rad).

## IP from THP-1 cells

For the immunoprecipitation of endogenous GSDMD, 4 mg VHH$_{GSDMD-1}$, VHH$_{GSDMD-2}$ or VHH$_{GSDMD-6}$ were covalently coupled to 0.25 g CNBr-activated Sepharose 6B (Sigma-Aldrich). $6.64 \times 10^7$ THP-1 cells were lysed in 10 ml lysis buffer (50 mM Tris pH 7.5, 500 mM NaCl, 1% NP40, Roche cOmplete™ Mini protease Inhibitor Cocktail) and cell debris was pelleted by centrifugation at 8000 g at 4 °C for 20 min. The lysate was added to the IP column and incubated on a shaker for 1 h at 4 °C. The column was washed with 200 ml wash buffer (50 mM Tris pH 7.5, 500 mM NaCl, 0.7% NP-40) and bound proteins were eluted with 500 μl elution buffer (0.2 M glycine pH 2.2). The pH of the eluted samples was neutralized by addition of 75 μl 1 M Tris pH 9.1. Samples were supplemented with a final concentration of 1X SDS-PAGE sample buffer and analyzed by SDS-PAGE followed by immunoblot or zinc staining.

## Zinc staining

Following SDS-PAGE, gels were briefly rinsed with ddH$_2$O and incubated with 0.2 M imidazole for 10 min on a shaker. Subsequently, gels were incubated with 0.3 M zinc acetate (ZnC$_4$H$_6$O$_4$) for 30 s while shaking vigorously. Gels were washed with ddH$_2$O and images were acquired using the Biorad Chemidoc system.

## LUMIER IP from HEK293T cells

HEK293T cells were seeded in a 6-well dish so that they reached 70–80% confluence the next day. Cells were transfected with 2.5 μg of pEXPR-vectors encoding either GSDMB-, GSDMD- or GSDME-Renilla luciferase using Lipofectamine™ 2000 transfection reagent (Thermo Fisher Scientific). High-binding Lumitrac 600 white 96-well plates (Greiner) were coated with 20 μg/mL of mouse anti-6xHis-Tag antibody (Thermo Fisher Scientific MA1-135) in PBS. The following day, cells were lysed lysis buffer (50 mM Hepes-KOH pH 7.9, 150 mM NaCl, 2 mM EDTA, 0.5% Triton X-100, 5% glycerol, Roche cOmplete™ Mini protease Inhibitor Cocktail) per well. Cell lysates were supplemented with 100 ng of recombinant VHH$_{GSDMD-1}$-LPETG-His, VHH$_{GSDMD-2}$-LPETG-His or VHH$_{GSDMD-6}$-HA-His and transferred to bound to the anti-His-coated Lumitrac 600 plate for three hours at 4 °C to immunoprecipitate (IP) VHH-His. After repeated washing with lysis buffer, Renilla luciferase substrate coelenterazine-h was added to the IP well or lysate controls. As positive control cells expressing the nucleoprotein of influenza A virus (NP-Renilla) and the respective nanobody (VHH$_{NP-1}$) were used. Luminescence was detected using a SpectraMax i3 instrument and the SoftMax Pro 6.3 Software (Molecular Devices). The IP values were normalized by the values of the lysate.

## Statistics and reproducibility

No statistical methods were used to predetermine sample size. The experiments were not randomized and the investigators were not blinded to allocation during experiments and outcome assessment.

## Reporting summary

Further information on research design is available in the Nature Portfolio Reporting Summary linked to this article.

## Data availability

The authors declare that the data supporting the findings of this study are available within the paper and its supplementary information files. Source data are provided with this paper. Structure coordinates and diffraction data of the human GSDMD–VHH$_{GSDMD-2}$–VHH$_{GSDMD-6}$ complex were deposited in the Protein Data Bank (http://www.pdb.org) under accession codes 7z1x. The coordinate data used in this study are available in the PDB database under accession codes 5ivo, 6n9o, 6vfe. Source data are provided with this paper.

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

## Acknowledgements

We would like to thank Jale Sariyar and Elif Tokmak for excellent technical assistance. The synchrotron data was collected at beamline P13 operated by EMBL Hamburg at the PETRA III storage ring (DESY, Hamburg, Germany). We would like to thank Isabel Bento for the assistance in using the beamline, and Raed Shalaby and Ana J Garcia-Saez, University of Cologne, for help with setting up the liposome leakage assays. M.G. and F.I.S. are funded by the Deutsche Forschungsgemeinschaft (DFG) under Germany's Excellence Strategy – EXC2151–390873048. F.I.S. is funded by DFG grant SFB1403-414786233. A.K. is supported by the DFG funded International Graduate School GRK 2168.

## Author contributions

A.K. expressed and purified proteins, performed biochemical experiments and crystallized the GSDMD complex, determined the structure with the help of G.H. I.J. supported the liposome leakage assay measurements, and J.M. performed SPR experiments. F.I.S. and L.D.J.S. identified VHHs. A.K., F.I.S. and M.G. conceptualized the study and A.K.

and M.G. wrote the manuscript. All authors contributed to the final version of the manuscript.

## Funding

## Competing interests

F.I.S. is cofounder and shareholder of Odyssey Therapeutics. The other authors declare no competing interests.
