## [Peer Review File · Nature Communications]

Pyroptosis inhibiting nanobodies block Gasdermin D pore formationReviewer #1 (Remarks to the Author):

In this study, Kopp et al. report on the identification of six nanobodies targeting full-length recombinant human GSDMD, a protein essential for pyroptosis and inflammation. These nanobodies have potential applications in dampening inflammation and furthering our understanding of inflammasome biology.

SPR analysis offered important information on the binding properties and epitopes of the nanobodies. VHHGSDMD-4 displayed a very low affinity ($\mu\text{M KD}$), while VHHGSDMD-1,2,3,5 demonstrated low nanomolar KDs for overlapping epitopes. VHHGSDMD-6 exhibited a higher KD (132 nM) for an epitope shared with VHHGSDMD-1,5 but not with VHHGSDMD-2,3.

Nanobodies were also tested for their ability to inhibit GSDMD-NTD-dependent (caspase-induced) nanopore formation in liposomes using an in vitro leakage assay. Despite sharing overlapping epitopes, only VHHGSDMD-1,2 demonstrated inhibitory effects, while VHHGSDMD-3,5 did not. VHHGSDMD-6 exhibited a partial, albeit significant, inhibitory effect.

The most interesting result of this work is the identification of two nanobodies (VHHGSDMD-1,2) binding with high affinity to GSDMD and showing a clear inhibitory effect on pore formation.

Crystal structures and superimposition analysis allowed the authors to suggest the molecular details of the inhibitory effect of VHHGSDMD-2, which might be bona-fide applied also to VHHGSDMD-1, since the two nanobodies bind to overlapping epitopes.

The discovery of these two nanobodies represents a relevant and original contribution in the field, providing the first GSDMD inhibitors a part the non-specific cysteine-reactive small molecules. Liposome leakage assay is the only "functional" test, but other functional effects (on pyroptosis and IL1B secretion) of cytosolically expressed nanobodies are apparently described in a companion paper (ref. 32).

Major points requiring consideration:

1) There appear some inconsistencies between SPR analysis and in vitro leakage assay. The SPR analysis reveals that VHHGSDMD-1,2,3,5 nanobodies bind with high affinity to overlapping epitopes, suggesting that they should affect GSDMD function similarly. However, when these nanobodies are tested in the in vitro leakage assay for their ability to prevent GSDMD-NTD-dependent pore formation, only VHHGSDMD-1,2 exhibit an inhibitory effect, while VHHGSDMD-3,5 do not.

This discrepancy between the binding properties and functional outcomes raises questions about the molecular mechanisms and potential reasons for the observed differences.

2) Limited structural data analysis. The manuscript reports the structural analysis of the binding interfaces only for two nanobodies, suggesting that VHHGSDMD-2 binds in the oligomerization interface of GSDMD-NTD (inhibiting oligomerization) whereas VHHGSDMD-6 binds on the top of the globular rim, (thus explaining the weak inhibition of oligomerization). However, both of these actually showed inhibitory effects on pore formation, and the partial but significant inhibition by VHHGSDMD-6 might be due to its lower affinity. Thus, the lack of structural data for the other nanobodies (VHHGSDMD-1,3,5), in particular the not inhibitory ones, limits the comprehensive understanding of their modes of action and how they influence GSDMD function. This information could help explain the inconsistencies mentioned in point 1 and provide a better rationale for the observed functional effects.

3) Unclear relevance of thermal stability data. The thermal stability data shows that all nanobodies except VHHGSDMD-5 increase the thermal stability of GSDMD. However, this increase is observed for both "neutralizing" (VHHGSDMD-1,2) and "not-neutralizing" (VHHGSDMD-3) nanobodies, as well as the partially neutralizing VHHGSDMD-6. Since the increased thermal stability is not exclusively associated with the neutralizing properties of the nanobodies, its relevance to their functional effects remains unclear. Further investigation is needed to determine if and how thermal stability data correlates with the functional outcomes of the nanobodies.

Other points:

4) VHHGSDMD-6's binding to immobilized GSDMD dissociates rapidly (almost entirely), as

demonstrated in Fig 1c, affecting the feasibility of conducting binning studies. In Extended Fig 2e, the second injection of all "secondary nanobodies," including VHHGSDMD-6, results in a binding signal, likely due to VHHGSDMD-6 from the first injection not being bound to GSDMD anymore. As a result, the data in Extended Fig 2e appears unconvincing and should be removed, along with its reference in Fig 1e.

5) Figures and the legends should be revised to increase their clarity: e.g., fig 1f; Extended Data Fig. 1-3; concentrations used (not just the ratio).

6) At line 99 (page 5), equilibrium dissociation constant should be indicated as K_D (kd usually identify dissociation rate constants). The authors could consider to report the association and dissociation rate constants. The 16-fold difference of the K_D between VHHGSDMD-1 and VHHGSDMD-2 is mainly due to the association rate constant, which can potentially explain the much lower difference (3-fold) observed in the liposome leakage assay.

7) It would be helpful to have a complete concentration-response curve for the effect of VHHGSDMD-6 in the leakage assay (as done for VHHGSDMD-1,2). This would help to better interpret the partial effect of this nanobody (see point 2).

8) The authors should also provide the effect of the nanobodies alone in the liposome leakage assay (Fig 2c, at 10 μ M)

9) Figures 2f and 2g are redundant.

Reviewer #2 (Remarks to the Author):

I have been reading this manuscript with great interest and I'm in principle in favour for publication with minor changes (see below) but there is one critical issue that has to be sorted out first. On lines 87 to 90, the authors lightly describe the identification of the nanobodies that are the subject of this study. And they refer to a manuscript that has been submitted: Schiffelers, L. D. J. et al. Antagonistic nanobodies reveal mechanism of GSDMD pore formation and unexpected therapeutic potential. What is really disturbing is that this second submitted manuscript has not been shared with the reviewers? However, comparing the titles of the submitted manuscript with the title of this manuscript under review indicates that there may be considerable overlap. Fig 1 of this manuscript is for sure also part of the submitted manuscript? Same for parts of figure 2? And the discussion in this manuscript is partially built on results that have been submitted to another journal (see lines 267-270)? It us appears that this manuscript is mirrored by another manuscript that we have not seen. I leave it to the editor to look into this and to take a decision regarding this issue because this relates to editorial policy.

Minor comments:

Line 14: Include Human as the first word. For an outstander, reading the current abstract, it is not clear if Gasdermin is an autologous human protein or an heterologous protein from a virus, bacterium or parasite , ...

Lines 30-35: Is GSDMD an extracellular protein? Or is it expressed and activated intracellularly? Again not clear from the first reading. I becomes clear if you read the discussion that the NTD is released in the cytoplasm to form a pore? Please make this evident allready in the introduction.

Line 72-75: Please refer to the original papers and not to a review of one of the authors of this manuscript only.

Line 87-90: See major comment above

Lines 267-270: It is strange that this part of the discussion is built on data that are submitted to

another journal? See major comment above.

Fig 3a: It is very difficult to visually discriminate the Nbs for the NTD in this overview figure. Wouldn't it be better to give the NTD in space filling and the Nb in a cartoon representation?

On 15 June, the editor provided me with a copy of the Schiffelers manuscript. After reading this paper, I came to the conclusion that both manuscripts have a different (non overlapping) focus. Accordingly, I recommend publication with minor revisions.

Jan Steyaert
www.steyaertlab.brussels

Reviewer #3 (Remarks to the Author):

The manuscript by Kopp et al. "Pyroptosis inhibiting nanobodies block Gasdermin D pore formation" present a set of nanobodies that can antagonistically target the terminal factor of the pyroptosis cascade GSDMD. This nice manuscript characterized the binding of this set of nanobodies and provided structural evidence to partially explain the mechanism of inhibition. Although the manuscript does not mechanistically advance our understanding of GSDMD's mechanism of regulation, these nanobodies could be a valuable tool for future mechanistic studies as well as for drug discovery screening. However, the manuscript does not present any functional assay to confirm that these nanobodies do indeed inhibit pyroptosis, as all the functional data are shown in the Schiffelers et al. manuscript, making Nature Communication a hard reach.

As the addition of functional assay won't likely be possible, the author should try to address more mechanistic insight into the mechanism of inhibition. The following few concerns should be addressed prior to publication.

1. The structure supports a model where VHH2 covers an oligomerization region preventing the formation of GSDMD pores. This can be considered an early inhibition method. I wonder if any of these nanobodies could destabilize a pre-formed pore reverting the pyroptosis state. The authors should generate full solubilized pores and in liposomes and incubate them with the nanobodies. This will add more mechanistic understanding in the absence of functional data.
2. "However, this class of inhibitors has a serious disadvantage: due to their cysteine reactivity, the compounds are not specific to GSDMD and binding to off-targets might cause unwanted side-effects in the human body". The authors make a correct point but should still try to address if the nanobodies have no specific interaction, particularly with other GSDMDs. Can the author perform a pull-down on cell lysate using recombinant nanobodies?
3. I would invite the author to rephrase the following: "Upon caspase cleavage, the NTD is released and undergoes large conformational changes that allow membrane binding, oligomerization, and pore formation" as cleavage of GSDMD has not been shown to induce immediate release of the NTD, the mechanism of membrane binding, oligomerization and pore formation are still debated in the field.
4. "In fact, we have re-engineered the first loop section 184-194 back into the GSDMD crystallization construct and reproducibly grown crystals that diffract up to 2.1 Å resolution, albeit without showing electron density for this section." Is this structure trimeric as the previous one? Does it still form this crystallization artifact? More information should be granted.
5. The biotinylation protocol for GSDMD is missing from the method.
6. "For the liposome leakage assay, 50 µl of the liposome solution, 0.5 µM GSDMD, 0.5 µM His-SUMO-caspase-4, and 0.5 µM VHH were mixed in a final volume of 200 µl in a dark-well glass

bottom plate and incubated at 37°C for 180 minutes. As control, the caspase inhibitor VX-765 was used at a concentration of 0.125 μM ." The author used a 4-fold less covalent Casp4 inhibitor (0.125 μM) than Casp4 (0.5 μM). How can Casp4 be fully inactive? Is it not common to use an excess inhibitor to prevent enzymatic activity?

7. Figure 2B The authors perform a liposome leakage assay using recombinant GSDMD and Casp4. However, in the material and methods is stated that the liposome leakage assay was performed using a GSDMD construct with a 3C cleavage site. The authors should properly address the mismatch. In addition, an SDS-PAGE of the proteins used for the assay would help show protein purity and proper digestion.

8. The CDR3 of VHHGSDMD-6 is particularly long comprising 15 residues and is stabilized by an additional disulfide bond between C100 and C110; a feature that contributes to the indistinguishable identification of the two nanobodies in the crystallographic electron density map (Extended Data Fig. 4b)." Did the author intend discernible?

9. The cryo-EM structure of the GSDMD pore was previously determined by the Wu lab" The lab name is not commonly used as a proper citation; please cite as Xia et al. 2021.

10. "Recombinant GSDMD (15 μM) was incubated with an equimolar amount of VHHGSDMD-1 or 435 VHHGSDMD-2 and caspase-4 (6 μM) at 37°C for 4 h. GSDMD cleavage by caspase-4 was analyzed by SDS-PAGE at the indicated time points." The authors use a high concentration of proteins for the assay in Fig.5A, I wonder why we cannot see Casp-4 from the gel. A control lane with only Casp-4 would be useful.

Detailed point-to-point reply to the Reviewers' comments:

Reviewer #1 (Remarks to the Author):

In this study, Kopp et al. report on the identification of six nanobodies targeting full-length recombinant human GSDMD, a protein essential for pyroptosis and inflammation. These nanobodies have potential applications in dampening inflammation and furthering our understanding of inflammasome biology.

SPR analysis offered important information on the binding properties and epitopes of the nanobodies. VHHGSDMD-4 displayed a very low affinity (μM KD), while VHHGSDMD-1,2,3,5 demonstrated low nanomolar KDs for overlapping epitopes. VHHGSDMD-6 exhibited a higher KD (132 nM) for an epitope shared with VHHGSDMD-1,5 but not with VHHGSDMD-2,3.

Nanobodies were also tested for their ability to inhibit GSDMD-NTD-dependent (caspase-induced) nanopore formation in liposomes using an in vitro leakage assay. Despite sharing overlapping epitopes, only VHHGSDMD-1,2 demonstrated inhibitory effects, while VHHGSDMD-3,5 did not. VHHGSDMD-6 exhibited a partial, albeit significant, inhibitory effect.

The most interesting result of this work is the identification of two nanobodies (VHHGSDMD-1,2) binding with high affinity to GSDMD and showing a clear inhibitory effect on pore formation. Crystal structures and superimposition analysis allowed the authors to suggest the molecular details of the inhibitory effect of VHHGSDMD-2, which might be bona-fide applied also to VHHGSDMD-1, since the two nanobodies bind to overlapping epitopes.

The discovery of these two nanobodies represents a relevant and original contribution in the field, providing the first GSDMD inhibitors a part the non-specific cysteine-reactive small molecules. Liposome leakage assay is the only “functional” test, but other functional effects (on pyroptosis and IL1B secretion) of cytosolically expressed nanobodies are apparently described in a companion paper (ref. 32).

Major points requiring consideration:

1) There appear some inconsistencies between SPR analysis and in vitro leakage assay. The SPR analysis reveals that VHHGSDMD-1,2,3,5 nanobodies bind with high affinity to overlapping epitopes, suggesting that they should affect GSDMD function similarly. However, when these nanobodies are tested in the in vitro leakage assay for their ability to prevent GSDMD-NTD-dependent pore formation, only VHHGSDMD-1,2 exhibit an inhibitory effect, while VHHGSDMD-3,5 do not. This discrepancy between the binding properties and functional outcomes raises questions about the molecular mechanisms and potential reasons for the observed differences.

We appreciate the Reviewer's comment. The observed disparity between the binding properties and functions of the nanobodies can be attributed to the experimental setup. We acknowledge that the terminology we initially used to describe the assay was imprecise and could potentially lead to misunderstandings of the data. Consequently, we have made adjustments to the terminology for the following reasons, which we will now outline.

The SPR-based epitope binning technique was used to characterize the binding sites of all six nanobodies on GSDMD. In this approach the nanobodies were tested pairwise for competitive binding to GSDMD. If both nanobodies tested in a pair bound to GSDMD, this unambiguously indicated that they bound to different epitopes. However, if the nanobodies interfered with each other's binding, there could be several explanations. It is possible that they bound to the exact same epitope, or their epitopes partially overlapped, causing steric hindrance that impeded the binding of the other nanobody. Additionally, the nanobodies might bind to different epitopes in proximity, leading to one nanobody blocking the binding of the other. Hence, to provide a more precise description, the term "mutually exclusive binding" should be used instead of "overlapping epitope." This clarification elucidates why nanobodies that exhibit mutually exclusive binding in the SPR epitope binning assay possess distinct functions. It is likely that their epitopes are separate but in close proximity, resulting in one nanobody sterically obstructing the binding of the second nanobody. We adjusted the text accordingly as following:

“Binding epitopes of the nanobodies on GSDMD were analyzed using an SPR-based epitope binning assay (Fig. 1d-f and Extended Data Fig. 2). Chemically biotinylated GSDMD was immobilized on an SPR sensor chip and nanobodies were injected as analytes in a pair-wise manner to investigate the possibility of mutually exclusive binding or simultaneous binding of both nanobodies to GSDMD (Fig. 1d). VHH_{GSDMD-1} and VHH_{GSDMD-5} exhibited mutually exclusive binding with all other nanobodies (Extended Data Fig. 2). VHH_{GSDMD-2} and VHH_{GSDMD-3} bound mutually exclusive, but for both nanobodies, additional binding of VHH_{GSDMD-6} was observed (Fig. 1e). According to these observations, VHH_{GSDMD-1} and VHH_{GSDMD-5}, as well as VHH_{GSDMD-2} and VHH_{GSDMD-3}, were grouped into one epitope bin, whereas VHH_{GSDMD-6} stands alone (Fig. 1f).”

2) Limited structural data analysis. The manuscript reports the structural analysis of the binding interfaces only for two nanobodies, suggesting that VHH_{GSDMD-2} binds in the oligomerization interface of GSDMD-NTD (inhibiting oligomerization) whereas VHH_{GSDMD-6} binds on the top of the globular rim, (thus explaining the weak inhibition of oligomerization). However, both of these actually showed inhibitory effects on pore formation, and the partial but significant inhibition by VHH_{GSDMD-6} might be due to its lower affinity. Thus, the lack of structural data for the other nanobodies (VHH_{GSDMD-1,3,5}), in particular the not inhibitory ones, limits the comprehensive understanding of their modes of action and how they influence GSDMD function. This information could help explain the inconsistencies mentioned in point 1 and provide a better rationale for the observed functional effects.

The availability of structural data is limited to VHH_{GSDMD-2} and VHH_{GSDMD-6}. Throughout the study, we made attempts to crystallize all six nanobodies in complex with GSDMD. However, successful crystallization was only achieved when a combination of VHH_{GSDMD-2} and VHH_{GSDMD-6} was used. In order to obtain structures of the remaining nanobodies in complex with GSDMD, alternative structural biology techniques such as cryo-EM could be explored, or modifications to the crystallization setup could be made, such as utilizing a different GSDMD construct. Nevertheless, it is important to note that alterations to the crystallization experiment do not guarantee the production of well-diffracting crystals suitable for structure determination.

3) Unclear relevance of thermal stability data. The thermal stability data shows that all nanobodies except VHH_{GSDMD-5} increase the thermal stability of GSDMD. However, this

increase is observed for both "neutralizing" (VHH_{GSDMD-1,2}) and "not-neutralizing" (VHH_{GSDMD-3}) nanobodies, as well as the partially neutralizing VHH_{GSDMD-6}. Since the increased thermal stability is not exclusively associated with the neutralizing properties of the nanobodies, its relevance to their functional effects remains unclear. Further investigation is needed to determine if and how thermal stability data correlates with the functional outcomes of the nanobodies.

We appreciate the Reviewer's comment regarding the consideration of a correlation between thermal stability and functional outcome of the nanobodies. Our view on the analysis of the melting temperature of the GSDMD–VHH complexes is instead shaped by the suitability of these nanobodies as potential chaperones for protein crystallization.

The GSDMD protein is largely unstructured with a long flexible segment that separates the NTD and CTD from each other and contains the caspase cleavage site; and the flexible loops in the cytosolic conformation of the NTD that transform into beta-strands when immersed in lipid membranes. In fact, 109 of 484 residues (~23%) appear unstructured in the cytosolic form of the protein (PDB ID 6n9o) and only deletion of two flexible segments (Δ 184-194/ Δ 247-272) led initially to successful crystallization attempts (Liu et al., 2019). We therefore considered the high degree of conformational flexibility of the protein an obstacle in the crystallization process, which we aimed to reduce through the stabilization with nanobodies. Nanobodies VHH_{GSDMD-1} and ₋₂ lead to a profound stabilization of GSDMD upon complex formation whereas VHH_{GSDMD-5} leads indeed lead to a lowering of the melting temperature in the complex with GSDMD. However, we do not consider that a stringent correlation between the change in thermal stability and the ability to inhibit pore formation exists for the nanobodies.

Other points:

4) VHH_{GSDMD-6}'s binding to immobilized GSDMD dissociates rapidly (almost entirely), as demonstrated in Fig 1c, affecting the feasibility of conducting binning studies. In Extended Fig 2e, the second injection of all "secondary nanobodies," including VHH_{GSDMD-6}, results in a binding signal, likely due to VHH_{GSDMD-6} from the first injection not being bound to GSDMD anymore. As a result, the data in Extended Fig 2e appears unconvincing and should be removed, along with its reference in Fig 1e.

Indeed, it is accurate to state that VHH_{GSDMD-6} exhibits rapid dissociation, as demonstrated in Figure 1c, and this characteristic does impact the epitope binning assay to small extent. It is also correct to assume that the second binding signal observed for VHH_{GSDMD-1} and VHH_{GSDMD-5} in Figure S2e (which appears small in comparison to the signal of VHH_{GSDMD-2} and ₋₃) most likely resulted from the continuous dissociation of VHH_{GSDMD-6} during the course of the measurement. However, it is important to note that a distinct disparity exists between the binding of VHH_{GSDMD-1/-5} and the additional binding observed for VHH_{GSDMD-2/-3}. This distinction, along with the supporting evidence from the sensorgrams shown figure S2b and S2c wherein the additional binding of VHH_{GSDMD-6} was observed upon the injection of VHH_{GSDMD-2} or ₋₃ first, solidifies the conclusion derived from the sensorgram presented in Figure S2e. It is also confirmed by the fact that the interaction matrix (Fig. 1e) is symmetric, which reflects that the interaction is associative. In other words, VHH_{GSDMD-6} binds on prebound GSDMD–VHH_{GSDMD-2} as VHH_{GSDMD-2} binds on prebound GSDMD–VHH_{GSDMD-6} complexes. Therefore, we like to retain the sensorgram in Figure S2e as it provides conclusive evidence on the nanobody–target protein interaction.

5) Figures and the legends should be revised to increase their clarity: e.g., fig 1f; Extended Data Fig. 1-3; concentrations used (not just the ratio).

We thank the Reviewer for this comment and have adapted the figure legends in the revised manuscript accordingly. The amended legends now read:

Fig. 1f, Binning of the nanobodies according to their properties in the competitive binding assay. Nanobodies VHH_{GSDMD-1} and -5 are grouped into one bin, as they showed mutually exclusive binding with all other nanobodies. VHH_{GSDMD-2} and -3 showed mutually exclusive binding to one another but allowed simultaneous binding of VHH_{GSDMD-6}, which itself does not share any binding similarities with the other nanobodies.

Extended Data Fig. 1 | Nanobodies and GSDMD variants used in this study. **a**, SEC elution chromatogram and **b**, SDS-PAGE of the recombinantly expressed and purified nanobodies VHH_{GSDMD-1} to VHH_{GSDMD-6}. **c**, SEC elution chromatogram and SDS-PAGE analysis of wild type, full length, human GSDMD (1-484). **d**, SEC elution chromatogram and SDS-PAGE analysis of a human GSDMD variant (1-484; residues 184-194 and 247-272 were deleted). **e**, SDS-PAGE of recombinant His-SUMO-Caspase-4 and western blot of His-SUMO-Caspase-4 using an anti-His-tag antibody.

Extended Data Fig. 2 | SPR-based epitope binning of VHH_{GSDMD} binding. **a**, Epitope binning assay with VHH_{GSDMD-1} submitted in a first titration step followed by a second titration step with one nanobody of the pool of five (VHH_{GSDMD-1}, -2, -3, -5, and -6). Chemically biotinylated human, full length GSDMD was immobilized on an SPR sensor chip and the competitive binding of nanobodies was tested in a pairwise manner. **b**, Epitope binning assay with VHH_{GSDMD-2} submitted in the first titration step. Association of the second nanobody to a distinct epitope can be observed for VHH_{GSDMD-6} as a second association event in the SPR sensorgram. **c-e**, Same as in **a** starting with VHH_{GSDMD-3}, -5, and -6, respectively.

Extended Data Fig. 3 | Effect of the nanobodies on the thermal stability of GSDMD. **a**, Melting temperature of human GSDMD at a concentration of 5 μ M. The melting temperature (T_m) was determined using nano-differential scanning fluorimetry (nanoDSF). **b**, Melting temperatures

of the six nanobodies alone at a concentration of 50 μM . **c-h**, Melting temperatures of the GSDMD–nanobody complexes at varying nanobody concentrations. For the titration experiment, 5 μM GSDMD was mixed with increasing concentration of the respective nanobody in a range of 1-50 μM . **i**, Summary of melting temperatures of GSDMD only, VHHs, and GSDMD–VHHs complexes at indicated concentrations.

6) At line 99 (page 5), equilibrium dissociation constant should be indicated as K_D (kd usually identify dissociation rate constants). The authors could consider to report the association and dissociation rate constants. The 16-fold difference of the K_D between VHH_{GSDMD-1} and VHH_{GSDMD-2} is mainly due to the association rate constant, which can potentially explain the much lower difference (3-fold) observed in the liposome leakage assay.

We fully agree with the Reviewer that the binding kinetics of the SPR measurements adds valuable information to the understanding and interpretation of the interaction. We therefore added a new Extended Data Table 1, listing the kinetic parameters and the dissociation constants. In addition, we thank the Reviewer of the attentive comment on the K_D designation and corrected its writing.

New Extended Data Table 1:

Binding to human GSDMD ^a	k_a (1/Ms)	k_d (1/s)	K_D (M)	Steady state affinity K_D (M)
VHH _{GSDMD-1}	6.48×10^6	1.98×10^{-3}	3.05×10^{-10}	5.47×10^{-10}
VHH _{GSDMD-2}	1.49×10^6	9.51×10^{-4}	6.37×10^{-10}	8.24×10^{-9}
VHH _{GSDMD-3}	2.80×10^6	1.54×10^{-3}	5.51×10^{-10}	2.19×10^{-9}
VHH _{GSDMD-4}	2.89×10^3	1.61×10^{-3}	5.59×10^{-7}	4.51×10^{-6}
VHH _{GSDMD-5}	3.02×10^6	1.49×10^{-3}	4.94×10^{-10}	4.14×10^{-9}
VHH _{GSDMD-6}	1.41×10^5	2.01×10^{-2}	1.43×10^{-7}	2.75×10^{-7}

^a Association and dissociation rate constants were determined at a flow rate of 30 $\mu\text{l}/\text{min}$. The association step was carried out for 120 s and the dissociation step for 300 s.

7) It would be helpful to have a complete concentration-response curve for the effect of VHH_{GSDMD-6} in the leakage assay (as done for VHH_{GSDMD-1,2}). This would help to better interpret the partial effect of this nanobody (see point 2).

We thank the Reviewer for pointing this out and now recorded dose-response curve measurements using the liposome leakage assay depending on the VHH_{GSDMD-6} concentration. The IC_{50} value of 1.3 μM is indeed six-fold weaker than those of VHH_{GSDMD-1} and the inhibitory effect also only partial. We have added the new dose-response curve for VHH_{GSDMD-6} in panel g of the revised Fig. 2.

8) The authors should also provide the effect of the nanobodies alone in the liposome leakage assay (Fig 2c, at 10 μM).

Yes, we fully agree with the Reviewer about the lack of this important control experiment. We have added the effect of the nanobodies alone in the liposome leakage assay in the new panel **d** of Fig. 2 in the revised manuscript. As can be seen from this control experiment, nanobodies VHH_{GSDMD-1} to VHH_{GSDMD-6} do not induce liposome leakage on their own compared to the induced pore formation by GSDMD and caspase-4.

9) Figures 2f and 2g are redundant.

This is correct and following the Reviewers' advice we deleted the previous Fig. 2g in the revised version of the manuscript.

Reviewer #2 (Remarks to the Author):

I have been reading this manuscript with great interest and I'm in principle in favour for publication with minor changes (see below) but there is one critical issue that has to be sorted out first. On lines 87 to 90, the authors lightly describe the identification of the nanobodies that are the subject of this study. And they refer to a manuscript that has been submitted: Schiffelers, L. D. J. et al. Antagonistic nanobodies reveal mechanism of GSDMD pore formation and unexpected therapeutic potential. What is really disturbing is that this second submitted manuscript has not been shared with the reviewers? However, comparing the titles of the submitted manuscript with the title of this manuscript under review indicates that there may be considerable overlap. Fig 1 of this manuscript is for sure also part of the submitted manuscript? Same for parts of figure 2? And the discussion in this manuscript is partially built on results that have been submitted to another journal (see lines 267-270)? It us appears that this manuscript is mirrored by another manuscript that we have not seen. I leave it to the editor to look into this and to take a decision regarding this issue because this relates to editorial policy.

Minor comments:

Line 14: Include Human as the first word. For an outstander, reading the current abstract, it is not clear if Gasdermin is an autologous human protein or an heterologous protein from a virus, bacterium or parasite , ...

We agree with the Reviewer and thank for this thoughtful comment. We have amended the Abstract accordingly to ensure that also a non-specialist reader will understand that GSDMD is a human endogenous protein.

“**Human** Gasdermin D (GSDMD) is a key mediator of pyroptosis, a pro-inflammatory form of cell death occurring downstream of inflammasome activation as part of the innate immune defence. Upon cleavage by inflammatory caspases **in the cytosol**, the N-terminal domain of GSDMD forms pores in the plasma membrane resulting in cytokine release and eventually cell death. Targeting GSDMD is an attractive way to dampen inflammation. In this study, six GSDMD targeting nanobodies were characterized in terms of their binding affinity, stability, and effect on GSDMD pore formation. Three of the nanobodies inhibited GSDMD pore formation in a liposome leakage assay, although caspase cleavage was not perturbed. We determined the crystal structure of human GSDMD in complex with two nanobodies at 1.9 Å resolution, providing detailed insights into the GSDMD–nanobody interactions and epitope binding. The pore formation is sterically blocked by one of the nanobodies that binds to the oligomerization interface of the N-terminal domain in the multi-subunit assembly. Our biochemical and structural findings provide new tools for studying inflammasome biology and build a framework for the design of novel GSDMD targeting drugs.”

Lines 30-35: Is GSDMD an extracellular protein? Or is it expressed and activated intracellularly? Again not clear from the first reading. It becomes clear if you read the

discussion that the NTD is released in the cytoplasm to form a pore? Please make this evident already in the introduction.

This point is well taken and we have adapted the text accordingly in the revised version of the manuscript.

The human gasdermin family consists of six differentially expressed members (GSDM A to E and PVJK/GSDMF) that exert diverse functions in inflammation and cell death^{1,2}. Gasdermin D (GSDMD) is a cytosolic protein that serves as a key mediator of pyroptosis, a pro-inflammatory form of cell death occurring in the context of microbial infection or tissue damage as part of the innate immune response^{3,4}. Sensing of cellular or pathogen-derived danger signals triggers the assembly of canonical and non-canonical inflammasomes, which leads to the activation of inflammatory caspases in the cytoplasm (caspase-1, -4, and -5 in human or caspases-1 and -11 in mice)^{5,6}. These caspases were found to cleave GSDMD at a conserved sequence motif (FLTD₂₇₅|GV in humans, LLSD₂₇₆|GI in mice), residing in a long linker region between the GSDMD N- and C-terminal domains of the 52.8 kDa protein^{3,4}. As in most gasdermins, the N-terminal domain (NTD) is cytotoxic and repressed by the auto-inhibitory C-terminal domain (CTD) in the inactive state⁷⁻¹⁰. Upon caspase cleavage, the NTD is released and undergoes large conformational changes that allow plasma membrane binding, oligomerization, and pore formation^{3,4,11,12}.

Line 72-75: Please refer to the original papers and not to a review of one of the authors of this manuscript only.

We agree with the Reviewer and cite now the following original papers (33-35):

33. Hamers-Casterman, C. *et al.* Naturally occurring antibodies devoid of light chains. *Nature* **363**, 446–448 (1993).
34. Dumoulin, M. *et al.* Single-domain antibody fragments with high conformational stability. *Protein Sci* **11**, 500–515 (2002).
35. Pardon, E. *et al.* A general protocol for the generation of Nanobodies for structural biology. *Nat Protoc* **9**, 674–693 (2014).

Line 87-90: See major comment above

We apologize for not having provided the full information on the accompanying manuscript that was uploaded back-to-back on bioRxiv. We now provide the full reference in the revised version of the manuscript.

GSDMD targeting nanobodies were raised by immunization of an alpaca with full-length recombinant human GSDMD protein. Identification by phage display and initial characterization of binding analyzed by enzyme-linked immunosorbent assay (ELISA) and LUMIER assays are described by Schiffelers *et al.*³⁷.

37. Lisa D. J. Schiffelers, Sabine Normann, Sophie C. Binder, Elena Hagelauer, Anja Kopp, Assaf Alon, Matthias Geyer, Hidde L. Ploegh, Florian I. Schmidt. Antagonistic nanobodies reveal mechanism of GSDMD pore formation and unexpected therapeutic potential.

Lines 267-270: It is strange that this part of the discussion is built on data that are submitted to another journal? See major comment above.

We agree with the Reviewer and deleted this sentence in the revised version of the manuscript. Instead, we now focus on the results of the biochemical characterization of the nanobodies and added a short discussion on the specificity to gasdermins, which we determined in the revision of the manuscript (new Extended Data Figures 4 and 5). In light of the comments mentioned above, we consider this more appropriate.

Fig 3a: It is very difficult to visually discriminate the Nbs for the NTD in this overview figure. Wouldn't it be better to give the NTD in space filling and the Nb in a cartoon representation?

We thank the Reviewer for this comment and adapted Fig. 3a in the revised manuscript accordingly.

On 15 June, the editor provided me with a copy of the Schiffelers manuscript. After reading this paper, I came to the conclusion that both manuscripts have a different (non overlapping) focus. Accordingly, I recommend publication with minor revisions.

Thank you very much for this consideration.

Jan Steyaert
www.steyaertlab.brussels

Reviewer #3 (Remarks to the Author):

The manuscript by Kopp et al. "Pyroptosis inhibiting nanobodies block Gasdermin D pore formation" present a set of nanobodies that can antagonistically target the terminal factor of the pyroptosis cascade GSDMD. This nice manuscript characterized the binding of this set of nanobodies and provided structural evidence to partially explain the mechanism of inhibition. Although the manuscript does not mechanistically advance our understanding of GSDMD's mechanism of regulation, these nanobodies could be a valuable tool for future mechanistic studies as well as for drug discovery screening. However, the manuscript does not present any functional assay to confirm that these nanobodies do indeed inhibit pyroptosis, as all the functional data are shown in the Schiffelers et al. manuscript, making Nature Communication a hard reach.

As the addition of functional assay won't likely be possible, the author should try to address more mechanistic insight into the mechanism of inhibition. The following few concerns should be addressed prior to publication.

1. The structure supports a model where VHH2 covers an oligomerization region preventing the formation of GSDM pores. This can be considered an early inhibition method. I wonder if any of these nanobodies could destabilize a pre-formed pore reverting the pyroptosis state. The authors should generate full solubilized pores and in liposomes and incubate them with the nanobodies. This will add more mechanistic understanding in the absence of functional data.

We understand the Reviewer's suggestion to analyse nanobody binding to a pre-formed, solubilized GSDMD pore in liposomes to see if any destabilization of the pore occurs. This is a very interesting question that will shed light on the mode of action of the nanobodies as inhibitors of pyroptosis. To our understanding, this evaluation requires single particle cryo-EM analyses to visualize the formation of the GSDMD pore in liposomes made from recombinant protein, following the application of nanobodies possibly at varying concentrations relative to the 33-subunit containing pore. The pre-formed pore of GSDMD has been reported by Xia et al., Nature (2021), and its structural evaluation must be considered as a great scientific achievement. Repeating these experiments in our study would require a large amount of time and man-power to set up this assay which we consider beyond the scope of the current manuscript. Moreover, the reported structure of the GSDMD pre-pore has a resolution of 6.9 Å, which is not sufficient to visualize structural details. As the administration of nanobody will possibly induce a collapse of the pore, the expectation for a highly ordered structure suitable for structure determination is rather low. Instead, a computational approach modelling the pore formation of GSDMD as recently reported by Schaefer and Hummer, eLife (2022), following the addition of nanobodies might be a suitable way to visualize the intervention of these nanobodies as pyroptosis inhibitors.

2. "However, this class of inhibitors has a serious disadvantage: due to their cysteine reactivity, the compounds are not specific to GSDMD and binding to off-targets might cause unwanted side-effects in the human body". The authors make a correct point but should still try to address if the nanobodies have no specific interaction, particularly with other GSDMs. Can the author perform a pull-down on cell lysate using recombinant nanobodies?

We thank the Reviewer for this comment and have performed further experiments to investigate the specificity of the nanobodies. THP-1 cells are widely used to study pyroptosis downstream of inflammasome activation (Zito, G. et al., *Int J Mol Sci* 2020; Yu, P. et al., *Signal Transduct Target Ther.* 2021). We investigated the expression of GSDMB, GSDMD and GSDME in THP-1 cells by western blot analysis. As positive controls, we used HEK293T cells transfected with the respective gasdermins as renilla-fusion proteins. Unfortunately, we were unable to express GSDMA-renilla and GSDMC-renilla and therefore did not examine these proteins. While we observed expression of GSDMD in THP-1 cells, none of the other gasdermins were expressed in this cell line. Therefore, we performed a pull-down assay using THP-1 cell lysate to test the overall specificity of the nanobodies, and transfected HEK293T cells to assess the cross-reactivity with the GSDMB and GSDME (Extended Data Fig. 5a).

To test the overall specificity of the nanobodies, we performed a pull-down assay utilizing THP-1 cell lysate and VHH_{GSDMD-1}, VHH_{GSDMD-2} and VHH_{GSDMD-6} immobilized on CNBr sepharose beads. The protein content of the samples at different stages of the pull-down was analysed using SDS-PAGE followed by zinc staining. As a control, we used resin that was not coupled to any nanobody. In the elution fractions of the nanobody coupled beads, a band corresponding to the molecular weight of GSDMD (52 kDa) was evident, which was absent in the control sample. Other protein bands observed in the elution fractions appeared to result from non-specific protein binding to the beads, as they were similarly observed in the control sample. The samples were further analysed by western blot using an antibody directed against GSDMD. The western blot revealed that all three nanobodies effectively pulled down endogenous GSDMD from THP-1 cells (Extended Data Fig. 5b).

To evaluate the potential cross-reactivity of the nanobodies with GSDMB and GSDME, we performed a pull-down using HEK293T cells transfected with plasmids encoding Gasdermin-renilla fusion proteins and recombinant nanobodies with a C-terminal His-tag. Although we were able to successfully express GSDMB-, GSDMD- and GSDME-renilla in this system, this was not possible for GSDMA-renilla and GSDMC-renilla (Extended Data Fig. 5a). Therefore, GSDMA and GSDMC were not included in the analyses. Cells expressing the respective gasdermins were lysed and subsequently combined with recombinant VHH_{GSDMD-1}, VHH_{GSDMD-2} or VHH_{GSDMD-6}. Following this, a pull-down was performed using a plate coated with an anti-His antibody directed against the His-tag fused to the nanobodies. As an indication for the efficiency of the pull-down, renilla luciferase activity resulting from the Gasdermin-renilla fusions was measured. As positive controls, a plasmid expressing the nucleoprotein of influenza A virus (NP-1) and a corresponding nanobody (VHH_{NP-1}) were used. VHH_{GSDMD-1}, and to a lesser extent also VHH_{GSDMD-2} and VHH_{GSDMD-6} were able to pull down GSDMD-renilla, but neither GSDMB- or GSDME-renilla.

Extended Data Fig. 5 | Specificity of VHH_{GSDMD-1}, VHH_{GSDMD-2}, and VHH_{GSDMD-6} to human GSDMD.

a, Western blot analyses of GSDMB, GSDMD and GSDME expression in THP-1 cells and HEK293T transfected with plasmids encoding the gasdermins as renilla-fusion proteins. **b**, A pull-down using CNBr beads coupled to VHH_{GSDMD-1}, VHH_{GSDMD-2} and VHH_{GSDMD-6} and THP-1 cell lysate was performed. The protein content of the samples at the different stages of the pull-down was analyzed using SDS-PAGE followed by zinc staining. The samples were further analyzed by western blot using an antibody directed against GSDMD. M: marker, TCL: total cell lysate, W: wash fraction, E1/E2: elution fraction 1/2. **c**, HEK293T cells were transfected with plasmids encoding GSDMB, GSDMD or GSDME as renilla-fusion proteins. Lysates of the transfected cells were combined with recombinant VHH_{GSDMD-1}, VHH_{GSDMD-2} or VHH_{GSDMD-6} harboring C-terminal His-tags. A pull-down using an anti-His antibody was performed and renilla luciferase activity was measured as readout for the efficiency of the pull-down. As positive controls a plasmid encoding nucleoprotein of influenza A virus (NP-1) and a corresponding nanobody (VHH_{NP-1}) were used. CL: cell lysate.

In addition, we analyzed the binding sites of the nanobodies VHH_{GSDMD-2} and VHH_{GSDMD-6} in a structure-based alignment of the sequences of all six human gasdermins. In fact, the sequence variability is very high in all gasdermins and there is no homology observed in the binding site of the nanobodies to any other protein sequence. This observation is in agreement with the high specificity of the nanobodies targeting only human GSDMD. The sequence evaluation is now shown in the new Extended Data Figure 5.

Extended Data Fig. 6 | Sequence alignment of human gasdermin proteins correlated to the secondary structure of GSDMD. Sequence alignment of human gasdermins GSDMA (UniProt ID Q96QA5), GSDMB (Q8TAX9), GSDMC (Q9BYG8), GSDMD (P57764), GSDME (O60443), and PJVK (Q0ZLH3) correlated to the crystal structure of the GSDMD–VHH_{GSDMD-2}–VHH_{GSDMD-6} complex determined here (PDB 7z1x). Secondary structure elements of GSDMD, the caspase cleavage site, the reactive Cys191 targetable by covalent inhibitors, and the N- and C-terminal domains are indicated at the top. Residues in red boxes are conserved in all gasdermins, similar residues are marked with red characters. The sequence alignment was performed with MultiAlin and the correlation to secondary structure elements and sequence conservation was done with ESPrpt 3.0 (ref. 59). Residues in GSDMD mediating direct interactions with VHH_{GSDMD-2} are boxed light and medium green according to the buried surface area (5–12 and >12 Å², respectively) as determined with PDBePISA (ref. 58). Similarly, residues in GSDMD mediating direct interactions with VHH_{GSDMD-6} are boxed light and medium blue. Hydrogen-bonds or salt bridges are indicated by a blue bar at the right side of the residue. The low degree of sequence conservation within the human gasdermin family is in agreement with the high specificity of the nanobodies, targeting only human GSDMD.

3. I would invite the author to rephrase the following: “Upon caspase cleavage, the NTD is released and undergoes large conformational changes that allow membrane binding, oligomerization, and pore formation” as cleavage of GSDMD has not been shown to induce immediate release of the NTD, the mechanism of membrane binding, oligomerization and pore formation are still debated in the field.

We agree with the Reviewer and rephrased the sentence the following:

Upon caspase cleavage, the two domains can dissociate by an as yet unknown mechanism, with the NTD capable of forming transmembrane pores. However, the exact mechanisms of NTD release, oligomerization, plasma membrane association, insertion, and conformational changes required to induced pore formation remain elusive^{3,4,11,12}.

4. "In fact, we have re-engineered the first loop section 184-194 back into the GSDMD crystallization construct and reproducibly grown crystals that diffract up to 2.1 Å resolution, albeit without showing electron density for this section." Is this structure trimeric as the previous one? Does it still form this crystallization artifact? More information should be granted.

We agree with the Reviewer that more information on this other protein construct would be valuable. In the revised version of the manuscript, we have modified this section the following:

In fact, we have re-engineered the first loop section 184-194 back into the GSDMD crystallization construct and reproducibly grown crystals that diffract up to 2.1 Å resolution *having the same space group and unit cell parameters*, albeit without showing electron density for this section. *Apart from that, the structure resembles the one described here and consists of a twisted dimer of a trimeric GSDMD–VHH_{GSDMD-2}–VHH_{GSDMD-6} complex.*

5. The biotinylating protocol for GSDMD is missing from the method.

We thank the Reviewer for this very attentive comment and added the protocol for the biotinylation of human GSDMD to the Methods section (lines 419–430). The biotinylation was carried out as follows:

For biotinylation, full length human GSDMD was transferred into modification buffer (100 mM NaH₂PO₄, 150 mM NaCl, pH 8.0) using Zeba™ spin desalting columns in accordance with the manufacturer's instructions. Biotinylation was performed using the ChromaLink™ Biotin Labeling kit from Solulink. Five equivalents of ChromaLink Biotin, dissolved in dimethylformamide (5 mg/ml), were added to the protein and incubated at room temperature for 90 minutes as per the manufacturer's recommendations. Any unreacted biotin was removed using Zeba™ Spin Desalting columns (ThermoFisher Scientific) that were pre-equilibrated with PBS. The protein in PBS was recovered, and the degree of labeling was assessed using a NanoDrop spectrophotometer

(ThermoFisher Scientific) by measuring the absorbance at 280 nm and 354 nm and employing the E1% ChromaLink Biotin molar substitution Calculator. For the SPR experiments, GSDMD protein labeled with approximately 2 biotins per molecule was utilized.

6. “For the liposome leakage assay, 50 μ l of the liposome solution, 0.5 μ M GSDMD, 0.5 μ M His-SUMO-caspase-4, and 0.5 μ M VHH were mixed in a final volume of 200 μ l in a dark-well glass bottom plate and incubated at 37°C for 180 minutes. As control, the caspase inhibitor VX-765 was used at a concentration of 0.125 μ M.” The author used a 4-fold less covalent Casp4 inhibitor (0.125 μ M) than Casp4 (0.5 μ M). How can Casp4 be fully inactive? Is it not common to use an excess inhibitor to prevent enzymatic activity?

We thank the reviewer for pointing this out. We determined the His-SUMO-caspase-4 concentration by measuring the absorption at A280 nm using a nanodrop spectrophotometer. As this method determines the amount of total protein in a sample, we overestimated the amount of His-SUMO-caspase-4 present in the sample, as our preparation is not homogenous. This can be observed in the Coomassie-stained SDS PAGE analysis of the affinity purification of caspase-4 in shown in Reviewer Fig. 1a (left panel). However, the presence of caspase-4 in the sample could be validated by western blot using an antibody directed against the His-tag of the fusion protein (Reviewer Fig. 1a, right panel). The wrongly estimated amount of 6 μ M His-SUMO-caspase-4 is nevertheless sufficient to cleave 15 μ M GSDMD over time course of 4 h (Reviewer Fig. 1b). A confirmation of the specificity of the Caspase-mediated cleavage reaction is given by the inhibition of the cleavage reaction by the VX-765 caspase inhibitor.

7. Figure 2B The authors perform a liposome leakage assay using recombinant GSDMD and Casp4. However, in the material and methods is stated that the liposome leakage assay was performed using a GSDMD construct with a 3C cleavage site. The authors should properly address the mismatch. In addition, an SDS-PAGE of the proteins used for the assay would help show protein purity and proper digestion.

We thank the Reviewer for pointing this mismatch out. This is indeed mistakenly described from us as we used indeed Caspase-4 for the cleavage reaction. We corrected this phrase in the Methods section of the revised manuscript. In addition, we now show a Coomassie-stained SDS PAGE analysis of the proteins used for the assay and the cleavage products of the substrate digestion in the Extended Data Fig. 1.

8. The CDR3 of VHH_{GSDMD-6} is particularly long comprising 15 residues and is stabilized by an additional disulfide bond between C100 and C110; a feature that contributes to the indistinguishable identification of the two nanobodies in the crystallographic electron density map (Extended Data Fig. 4b).” Did the author intend discernible?

Yes, fully correct! Thank you very much for this advice. In fact, for more than half a year we had a diffraction map at 3.0 Å resolution only for the tripartite GSDMD–VHH_{GSDMD-2}–VHH_{GSDMD-6} complex. At this point, the disulfide bond in the CDR3 of VHH_{GSDMD-6} contributed to the clear distinction between the two nanobodies as it was identified already at low resolution. We corrected the sentence accordingly.

9. The cryo-EM structure of the GSDMD pore was previously determined by the Wu lab” The lab name is not commonly used as a proper citation; please cite as Xia et al. 2021.

Corrected.

10. “Recombinant GSDMD (15 μM) was incubated with an equimolar amount of VHH_{GSDMD-1} or VHH_{GSDMD-2} and caspase-4 (6 μM) at 37°C for 4 h. GSDMD cleavage by caspase-4 was analyzed by SDS-PAGE at the indicated time points.” The authors use a high concentration of proteins for the assay in Fig.5A, I wonder why we cannot see Casp-4 from the gel. A control lane with only Casp-4 would be useful.

We thank the Reviewer for pointing this out and agree, that a control lane showing Casp-4 is useful. We now added a Coomassie-stained SDS PAGE analysis of use purified Casp-4 protein and a western blot of the protein in the Extended Data Fig. 1e. In addition, we corrected the sentence regarding the high concentration of the cleavage enzyme estimating the lower concentrations as described in point 6 of this review. Due to the low concentration of the Casp-4 used, a band of the protein is not seen in the Coomassie stained gel.

We thank all Reviewers for their considerate comments and kind assessment of our study.

Reviewer #1 (Remarks to the Author):

The authors addressed in a satisfactory manner all the points raised previously.

It remains just one point to be considered, related to the new Extended Data Table 1. In this table, two values of KD are given for each nanobody, one calculated as the ratio k_d/k_a and one obtained as "Steady state" KD. Notably, for 4 nanobodies the two values differ markedly, by one order of magnitude.

In their work, the authors mainly consider the "steady state" KD (reported in Fig 1) but, to this reviewer, the most reliable KD values are those estimated by the k_d/k_a ratio.

The authors have not been described how the "steady state" KDs were obtained, but from the visual inspection of the sensorgrams in Fig.1 it appears that no steady state is reached in most of them, raising some doubts on the validity of these KDs.

If there are no convincing explanations to support the choice of the "steady state" KDs, the authors should report and consider only the KD values estimated by k_d/k_a ratio, in Extended Table 1 and in Fig.1. This should also be considered in the text of Results and Discussion.

Reviewer #2 (Remarks to the Author):

No further comments. Excellent paper.

Jan Steyaert

Reviewer #3 (Remarks to the Author):

The manuscript by Kopp et al. "Pyroptosis inhibiting nanobodies block Gasdermin D pore formation" was nicely made to begin with. I am satisfied that the author addressed my concerns, and happy to recommend it for publication.

Detailed point-to-point reply to the Reviewers' comments:

Reviewer #1 (Remarks to the Author):

The authors addressed in a satisfactory manner all the points raised previously.

It remains just one point to be considered, related to the new Extended Data Table 1. In this table, two values of K_D are given for each nanobody, one calculated as the ratio k_d/k_a and one obtained as "Steady state" K_D . Notably, for 4 nanobodies the two values differ markedly, by one order of magnitude. In their work, the authors mainly consider the "steady state" K_D (reported in Fig 1) but, to this reviewer, the most reliable K_D values are those estimated by the k_d/k_a ratio.

The authors have not been described how the "steady state" K_D s were obtained, but from the visual inspection of the sensorgrams in Fig.1 it appears that no steady state is reached in most of them, raising some doubts on the validity of these K_D s.

If there are no convincing explanations to support the choice of the "steady state" K_D s, the authors should report and consider only the K_D values estimated by k_d/k_a ratio, in Extended Table 1 and in Fig.1. This should also be considered in the text of Results and Discussion.

We understand the Reviewers' concern and agree that the parameters derived from the kinetic association and dissociation rate constants are more accurate, particularly as steady state levels in the time course of the SPR experiments (120 s association phase and 300 s dissociation phase at a flow rate of 30 μ l/min) are not necessarily reached (see for example the association process of VHH_{GSDMD-4}). We therefore changed the K_D parameters provided in the main text and in Figure 1c to the values derived from the kinetic measurements (k_d/k_a).

Reviewer #2 (Remarks to the Author):

No further comments. Excellent paper.

Jan Steyaert

Reviewer #3 (Remarks to the Author):

The manuscript by Kopp et al. "Pyroptosis inhibiting nanobodies block Gasdermin D pore formation" was nicely made to begin with. I am satisfied that the author addressed my concerns, and happy to recommend it for publication.

We thank all Reviewers for a thoughtful and highly constructive review process.